# Arctic freshwater outflow suppressed Nordic Seas overturning and oceanic heat transport during the Last Interglacial

Mohamed M. Ezat [1] ✉, Kirsten Fahl [2] & Tine L. Rasmussen[3]

The Last Interglacial period (LIG) was characterized by a long-term Arctic atmospheric warming above the preindustrial level. The LIG thus provides a case study of Arctic feedback mechanisms of the cryosphere-ocean circulation-climate system under warm climatic conditions. Previous studies suggested a delay in the LIG peak warming in the North Atlantic compared to the Southern Ocean and evoked the possibility of southward extension of Arctic sea ice to the southern Norwegian Sea during the early LIG. Here we compile new and published proxy data on past changes in sea ice distribution, sea surface temperature and salinity, deep ocean convection, and meltwater sources based on well-dated records from the Norwegian Sea. Our data suggest that southward outflow of Arctic freshwater supressed Nordic Seas deep-water formation and northward oceanic heat transport during the early LIG. These findings showcase the complex feedback interactions between a warming climate, sea ice, ocean circulation and regional climate.

The Arctic cryosphere is transforming rapidly in response to ongoing climate change with profound implications for regional and global climate, future sea level rise and stability of ecosystems[1,2]. For example, it has been suggested that positive heat and freshwater flux anomalies in the Arctic are the primary cause of the suggested recent slow-down of the Atlantic Meridional Overturning Circulation (AMOC), a crucial regulator of the earth's climate and fundamental for the mild climate of northwest Europe[3]. This has also been linked to the so called sub-polar North Atlantic "Warming Hole" where the ongoing warming is slower than elsewhere on the globe or has even cooled down over recent years[3]. A conceptual explanation of these effects is that the associated northward oceanic meridional transport of heat and salt within the AMOC is balanced by a southward flow of cold deep water that is mainly formed in the Nordic Seas (the Greenland, Iceland, and Norwegian Seas) and the subpolar gyre. Thus, a southward advection of Arctic positive buoyancy (heat and freshwater) anomalies may have suppressed the deep-water formation in these key areas and thus altering the AMOC and the associated northward oceanic heat transport. However, there is no agreement regarding the relative role of

individual components of Arctic climate change (e.g., melting of the Greenland Ice Sheet, sea ice reduction-induced freshwater and heat flux anomalies) to the suggested slowdown of the AMOC (refs. 3–5) or the associated changes in deep ocean convection[6–8]. A broader paleoclimatic perspective, permitting a wider range of boundary conditions and different rates of climate change to be investigated, is a key for understanding the interplay and feedback mechanisms between changes in the Arctic cryosphere, global ocean- and atmosphere circulation, and climate.

The Last Interglacial period (LIG; ~128–117 ka, thousands of years before present) was characterized by a warmer-than-present global climate, a smaller ice volume and a higher sea level[9]. It also provides a case study of long-term polar atmospheric warming above the pre-industrial level[10,11] and the response of the Arctic climate system and strength of the deep ocean convection in the Nordic Seas to this warming. However, proxy studies gave conflicting results on changes in Arctic Ocean sea ice cover during the LIG for the central Arctic Ocean (comprising the Eurasian and Canadian Basins) [12–17]. For example, sea ice biomarker proxies suggest the existence of a permanent

[1]iC3 - Centre for ice, Cryosphere, Carbon and Climate, Department of Geosciences, UiT, The Arctic University of Norway, Tromsø, Norway. [2]Alfred Wegener Institute Helmholtz Centre for Polar and Marine Research, Am Handelshafen 12, Bremerhaven, Germany. [3]Department of Geosciences, UiT, The Arctic University of Norway, Tromsø, Norway. ✉e-mail: mohamed.ezat@uit.no

sea ice cover all year round in the central Arctic Ocean during the LIG[12], while the occurrence of sub-polar planktic foraminifera in sediments assigned to the LIG from the central Arctic argue for sea ice free summers[13]. Further, recent studies have questioned the LIG age of these records from the central Arctic Ocean[18,19]. Also, because of the extremely low sedimentation rates in the central Arctic Ocean, the temporal resolution and age uncertainties in these marine proxy records do not allow for detailed reconstructions of the development of oceanographic changes e.g., due to varying insolation forcing across the LIG. Studies from the central and northern Nordic Seas have mostly focused on identifying the LIG climate optimum and comparing with

**Fig. 1 | Map of Nordic Seas and proxy records for the peak Marine Isotope Stage (MIS) 6, penultimate deglaciation, MIS 5e comprising the last interglacial, and glacial inception MIS 5d. a** Map showing major surface (red and white arrows) and bottom (black arrows) currents in the northern North Atlantic and Nordic Seas[69]. Red and white arrows indicate the northward Atlantic surface water inflow and southward polar water outflow, respectively. EGC and EIC refer to the East Greenland Current and East Icelandic Current, respectively. Circles highlight the location of the southern Norwegian Sea sediment cores JM11-FI-19PC (black, 1179 m water depth), MD95-2009 (brown, 1217 m water depth) and LINK16 (blue, 773 m water depth) and the North Atlantic sediment core ENAM33 (purple circle, 1217 m water depth). The map is modified after ref. 70. **b** Benthic foraminiferal $\delta^{18}$O from cores JM11-FI-19PC (black, ref. 24), LINK16 (blue, this study), MD95-2009 (brown, ref. 23). **c** Planktic foraminiferal $\delta^{18}$O from cores JM11-FI-19PC (black, ref. 24), LINK16 (blue, this study), MD95-2009 (brown, ref. 23). **d** Percentage of *Neogloboquadrina pachyderma* from core LINK16 (blue, ref. 31) and core MD95-2009 (brown, ref. 23). **e** Percentage of planktic foraminiferal species *N. pachyderma* from core ENAM33 (purple, ref. 23). **f** Summer solar insolation at 60° N (solid line) and 90° N (dashed line) (ref. 45). The vertical black line marks tephra layer 5e-Low/BasIV present in the three sediment cores from the southern Norwegian Sea as well as in the North Atlantic sediment core ENAM33. LIG stands for the Last Interglacial.

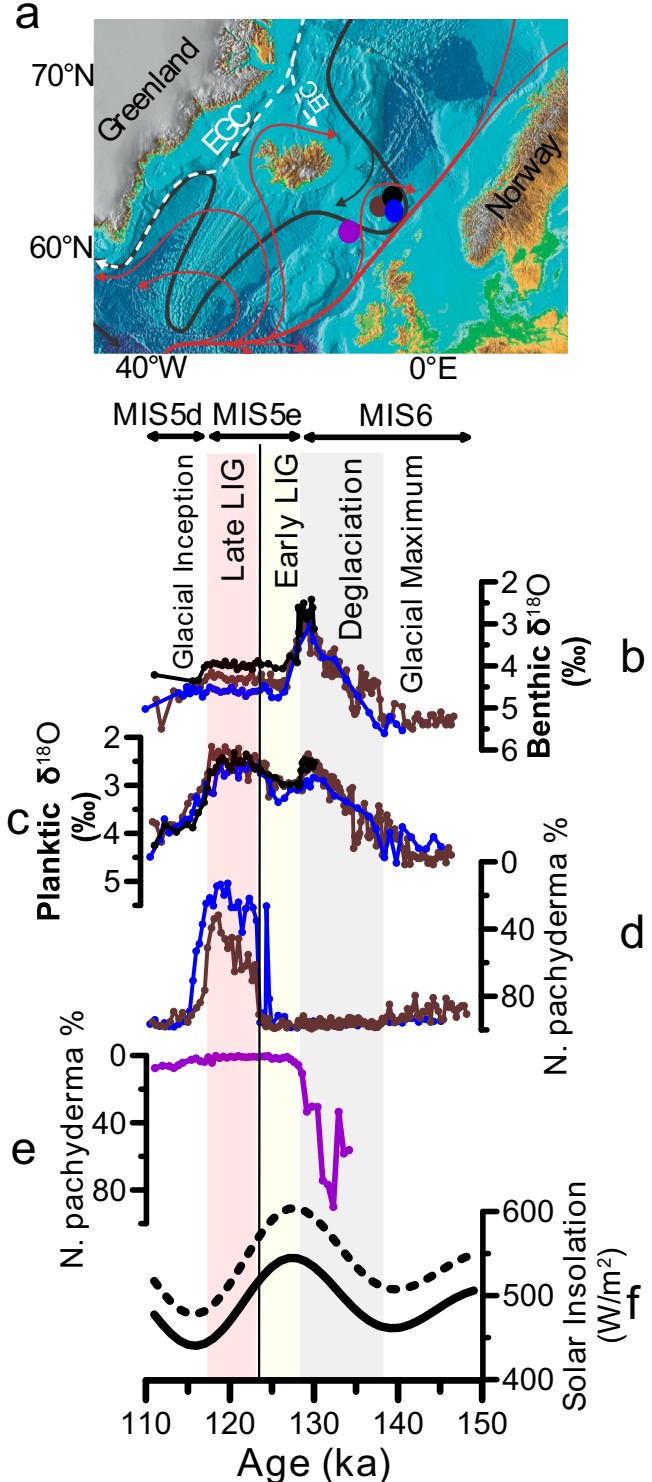

the Holocene climate state or variability, and suggested sea ice free conditions all year round during the LIG peak warming[17,20–22]. Higher resolution and well-dated records from the southern Norwegian Sea and the subpolar North Atlantic suggested a delay in the LIG peak warming in the North Atlantic compared to the Southern Ocean[23–26]. Further, planktic foraminiferal assemblages from the southern Norwegian Sea suggest summer temperatures below 5 °C (for example, ref. 26 based on sediment core MD95-2009, see Fig. 1a for core location), which raises the possibility that winter sea ice may have expanded much further south in the Nordic Seas during a globally warmer-than-present time interval compared to the historical and preindustrial periods. The early LIG cooling in the southern Norwegian Sea is also supported by dinocyst assemblage analyses in core MD95-2009 (ref. 27). Although the early LIG cooling was attributed to icesheet melting and suppression of Nordic Seas deep convection[25,27,28], neither the sources of meltwater nor deep water formation processes have been constrained by proxy data. Nevertheless, a more recent study based on diatom assemblages in the nearby core JM-FI-19PC suggested that the sea surface during the early LIG was warm[29] (see Fig. 1a for core location), but a detailed comparison between these sea surface temperature proxies (i.e., diatom, dinocyst, and planktic foraminifera) is still lacking. Also, a more direct proxy for sea ice such as the biomarker IP$_{25}$ (a C$_{25}$ Isoprenoid Lipid; ref. 30) has not yet been applied on the Norwegian Sea sediment records that record the early LIG cooling.

Ultimately, the question of the early LIG cooling in the Norwegian Sea and potential responsible processes remains an open debate. In this study, we utilize a multi-proxy approach (diatom, dinocyst, and planktic foraminiferal assemblages, sea ice biomarkers, planktic foraminiferal Na/Ca and Ba/Ca, and benthic foraminiferal assemblages) to reconstruct the development of sea ice, sea surface temperature, deep ocean convection as well as changes in freshwater input and their sources during the LIG in the Norwegian Sea. We focus on three sediment cores retrieved from the southern Norwegian Sea (Fig. 1a) because they can uniquely be compared with high confidence to their counterparts from the North Atlantic Ocean due to the consistent presence of a chronostratigraphic tephra layer[23,24,31] dated to 123.7 ka (ref. 32, Fig. 1; see "Methods"). Figure (1b, c) shows that the three sediment cores share the same characteristic patterns in oxygen isotope ratios ($\delta^{18}$O) measured in planktic (surface ocean dwellers) and benthic (seafloor dwellers) foraminifera across the penultimate glacial maximum, penultimate deglaciation, the LIG and the last glacial inception of marine isotope stages (MIS) 6 to MIS 5d, providing confidence in comparing the different types of proxy data between the three records.

## Results and Discussions

### Sea surface temperature development during the LIG

Faunal and floral assemblages including planktic foraminifera, dino-flagellate cysts, and diatoms have been widely used to reconstruct past changes in surface ocean conditions[29,32–34]. Planktic foraminiferal assemblage studies from the southern Norwegian Sea found that during the penultimate deglaciation (-138–128 ka) and the early LIG (-128–123.5 ka), the polar species *Neogloboquadrina pachyderma* constituted -90–100% of the planktic foraminiferal assemblages (ref. 23, Fig. 1d) suggesting that summer sea surface temperatures were below 5 °C (ref. 26). Diatom assemblages[29] when placed on similar age scales as the planktic foraminiferal data (see "Methods") also suggest a delay in the LIG warming peak and that Holocene-like SST only established at -123.5 ka (Figs. 2e, f; 3a). This is also supported by the dinocyst assemblages as the warm Atlantic water indicator species such as *Operculodinium centrocarpum* significantly increased in abundance only at -124 ka, whereas the early LIG was dominated by the colder water species *Brigantedinium* spp. (ref. 27, see Fig. 2g). Further, a close inspection of the planktic foraminiferal and diatom assem-blages (Fig. 3a, b) reveals three distinct phases of sea surface tem-perature development during the LIG namely at -128–126.5 ka, -126.5–123.5 ka, and -123.5–117 ka. During the earliest LIG (-128–126.5) both diatom and planktic foraminiferal assemblages were dominated by cold species, similar to their compositions during terminations II (Fig. 2e, f, g). The assemblage compositions of the earliest LIG are also similar to the assemblage compositions during the cold glacial stadial periods[23,29] when the Norwegian Sea was covered by perennial or seasonal sea ice[35]. This suggests that summer sea surface temperatures were below -5 °C during both the latest part of penultimate deglacia-tion and earliest LIG and raises the possibility that winter sea ice may have persisted during the earliest LIG—a possibility that we will inves-tigate latter.

During the second phase of the LIG (-126.5–123.5 ka), the cold-water diatom species were gradually being replaced by warm diatom species (Fig. 3a), but the planktic foraminiferal assemblages remained dominated by the polar species *N. pachyderma* similar to the earliest LIG (Fig. 3b). Yet, both assemblages indicate colder surface ocean compared to the pre-industrial and the entire interval of the Holocene (Figs. 3a, b). Below -5 °C, the planktic foraminiferal assemblages are dominated by *N. pachyderma* (-90–100%) (e.g., ref. 32) and thus changes in temperature from subfreezing temperature to -5 °C may not be recorded. Similar discrepancies between planktic foraminiferal and diatom assemblages were also previously related to changes in the thermal structure in the upper water column as diatoms reflect surface water conditions (0–50 m) and planktic foraminifera record subsur-face waters (e.g., ref. 36). Additionally, the planktic foraminiferal assemblages could be partly affected by dissolution during this inter-val in which less abundant and more dissolution prone subpolar planktic species are more affected as outlined by Zamelczyk et al. (ref. 37) for the last glacial period. Further, we observe a remarkable short-term decrease in the % *N. pachyderma* at 124.5 ka in only one sediment core (ref. 31, Fig. 2f). Although this may suggest a rapid and large amplitude change in sea surface temperature, a careful com-parison with sedimentological indicators suggest that the increase in the subpolar species is due to bioturbation and thus not reflecting temperature change (presence of an ash pod indicates this is the case; c.f., ref. 38, Supplementary Fig. 1). Overall, summer sea surface tem-perature during this time interval (-126.5–123.5 ka) was increasing but was lower than the pre-industrial and Holocene range.

The third LIG phase is the late LIG -123.5–117 ka when the per-centage of warmer-water indicating planktic foraminiferal and diatom species reached similar or higher values compared to the Holocene (Fig. 3a, b). Interestingly, a recent study from the Labrador Sea also suggested three LIG phases particularly the earliest LIG (128–126.5 ka) when the Labrador Sea was seasonally covered by sea ice and that

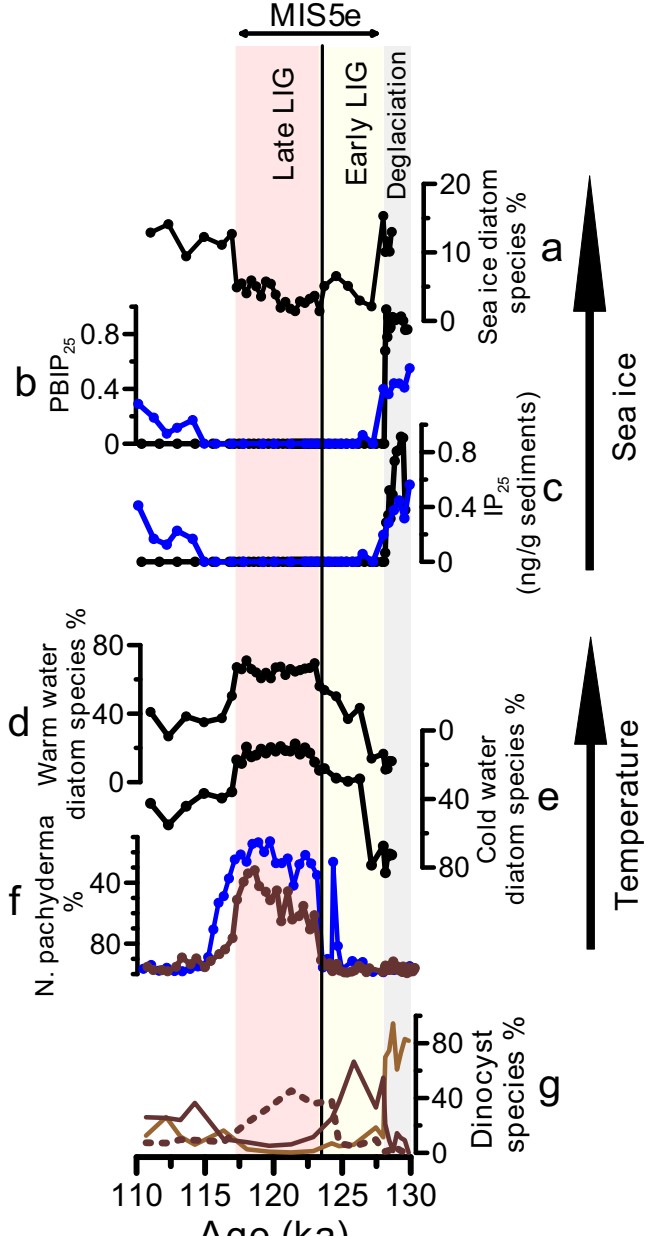

**Fig. 2 | Development of sea surface temperature and sea ice in the Norwegian Sea during the penultimate deglaciation and Last Interglacial (LIG).** Sea ice proxies: (**a**) Relative abundance of sea ice indicating diatom species[29]; (**b**) Sea ice index $P_BIP_{25}$ from two sediment cores (this study); and (**c**) Concentration of $IP_{25}$ (a $C_{25}$ Isoprenoid Lipid) from two sediment cores (this study). Temperature proxies: (**d**) Relative abundance of warm water-indicating diatom species[29]; (**e**) Relative abundance of cold water-indicating diatom species[29]; and (**f**) Relative abundance of the polar planktic foraminiferal species *Neogloboquadrina pachyderma*[23,31]. **g** Relative abundance of the key dinocyst species *O. centrocarpum* (dashed line), *Brigantedinium* spp. (dark brown) and *Islandinium minutum* (light brown)[27]. Black, blue and brown colors refer to data from sediment core JM11-FI-19PC, LINK16 and MD95-2009, respectively (see Fig. 1a for the core locations). MIS refers to Marine Isotope Stage. The vertical black line marks tephra layer 5e-Low/BasIV present in the three sediment cores JM11-FI-19PC, LINK16 and MD95-2009.

persistent inflow of warm water only occurred at 124–116 ka (ref. 39). In conclusion, planktic foraminiferal and diatom assemblages collec-tively suggest: 1) Norwegian Sea summer sea surface temperature did not change significantly between Termination II and the earliest last interglacial (-128–126.5 ka) with temperature < 5 °C; 2) a transitional

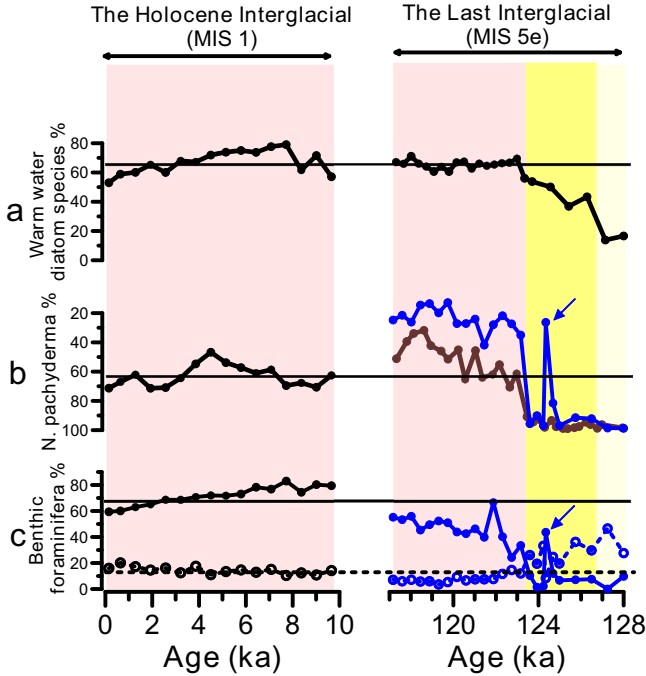

**Fig. 3 | Comparison of surface and deep ocean conditions during the Last Interglacial (128–117 ka; right) and the Holocene (10–0 ka; left) at the studied core sites. a** Relative abundance of warm-water indicating diatom species[29]. **b** Relative abundance of the polar planktic foraminiferal species *Neogloboquadrina pachyderma*[23,31]. **c** Relative abundance of the benthic foraminiferal species *Cassidulina neoteretis + Cassidulina reniforme* (solid line and filled circles) and *Melonis barleeanus* (dashed line and open circles). Black, blue and brown colors refer to data from sediment cores JM11-FI-19PC, LINK16 and MD95-2009, respectively (see Fig. 1a for core locations). Horizonal lines refer to the average Holocene values. Blue arrows refer to an interval in core LINK16 that is likely impacted by bioturbation (see supplementary Fig. 1). MIS and LIG stand for Marine Isotope Stage and the Last Interglacial, respectively.

period (~126.5–123.5 ka) with colder-than-present, but increasing, summer sea surface temperature; and 3) persistent warm Atlantic inflow and similar water column thermal structure to present-day during 123.5–117 ka.

### Sea ice distribution and seasonality during the LIG

Did winter sea ice expand to the southern Norwegian Sea during the globally warmer-, but regionally colder-than-present early LIG (~128–123.5 ka), in particular during the coldest interval 128–126.5 ka? To answer this question, we measured the biomarker $IP_{25}$ and sterol concentrations in two of the three studied sediment cores. The biomarker $IP_{25}$ is biosynthesized by a few diatom species that live in sea ice[40], and thus its presence is an indicator of seasonal sea ice[30]. Complete absence of $IP_{25}$ indicates either permanent sea ice (because of the too limited light and nutrient availability) or year-round open ocean conditions (due to the absence of ice algae). In such case, the additional view on phytoplankton biomarkers such as brassicasterol and dinosterol can distinguish between the permanent sea ice versus open ocean conditions. Taken in consideration both $IP_{25}$ and sterol concentrations, sea ice indices $P_BIP_{25}$ and $P_DIP_{25}$ were suggested as semi-quantitative indicators of sea ice changes on a scale that ranges from zero (open ocean conditions) to 1 (perennial sea ice) (ref. 41).

High $IP_{25}$ concentrations, low brassicasterol concentrations and sea ice index values of ~0.4–0.8 during the penultimate deglaciation (~130–128 ka) suggest seasonal sea ice conditions (Fig. 2b, c). This is further supported by the high abundance (~60–90%) of the dincoyst species *Islandinium minutum* (ref. 27, see Fig. 2g), which may indicate

prolonged periods of sea ice coverage[42]. During the entire LIG, $IP_{25}$ is absent, the concentrations of brassicasterol are at maximum and the sea ice index values are zero (Fig. 2; Supplementary Fig. 2) indicating open water conditions all year round. Overall, the sea ice development from the penultimate deglaciation and across the LIG is also in agreement with the sea ice related diatom species and dinocyst assemblages (refs. 27,29, Fig. 2; Supplementary Fig. 2). After the end of the LIG, the biomarker ($IP_{25}$, $P_BIP_{25}$) data and sea ice-indicating diatom species suggest differences in the timing of the appearance of sea ice. The abundance of sea ice indicating diatom species starts to increase (close to the penultimate deglaciation levels) immediately at the end of LIG at ~116.5 ka (Fig. 2b), whereas $IP_{25}$ was absent until ~114 ka (Fig. 2c, d). It could be that winter sea ice started to extend southward at ~116.5 ka, but it only reached the Faroe Islands margin at 114 ka or later – such a prolonged open ocean corridor during the last glacial inception may have provided moisture for ice sheet growth[24,43,44].

### Constraints on the sources of potential freshwater anomalies and implications

What caused the regional cooling during the early LIG? The early LIG interval (~128–124 ka) was characterized by high summer solar insolation at high northern latitudes – significantly higher than pre-industrial levels (ref. 45, Fig. 1f). Previous studies suggested the potential cooling of the early LIG may have been caused by persistent melting of ice sheets[25,27,28] that could have disturbed open ocean convection in the Nordic Seas and the subpolar North Atlantic and thus suppressed the northward oceanic heat transport. Yet, proxy evidence for an early LIG meltwater event (i.e., a decrease in salinity) and its source is not well-constrained. To date, it has proven difficult to develop a reliable independent proxy for sea surface salinity. The Na/Ca in planktic foraminifera is proposed as a proxy for surface ocean salinity, although non-salinity parameters may also play a role[46,47]. Using Na/Ca measured in shells of *N. pachyderma* as a qualitative indicator for sea surface salinity, it suggests lower salinity during the early LIG compared to the late LIG (Fig. 4a, b). This is also supported with the dinocyst assemblages that are dominated by *Brigantedinium* spp. (mostly *Brigantedinium simplex*) during the early LIG (ref. 27, see Figs. 2g, 4a), which likely indicate lower salinities and temperatures[48,49]—compared with the late LIG, which is dominated by warmer water species such as *O. centrocarpum* (ref. 27 Fig. 2g).

What was the source of the freshwater anomaly in the southern Norwegian Sea during the early LIG? Constraining the freshwater sources during the early LIG could provide invaluable insights on cryosphere-ocean circulation-climate interactions under interglacial conditions. Potential freshwater sources include glacial runoff (and melting of icebergs), increased river input to the region, local precipitation, meltwater from sea ice, and/or a decrease in the inflow of northward Atlantic surface water. The early LIG in the southern Norwegian Sea is characterized by relatively high planktic foraminiferal $\delta^{18}O$ and seawater $\delta^{18}O$ (Fig. 4f; ref. 24) as well as a complete absence of Ice Rafted Debris (IRD) (Fig. 4d), which do not support a freshwater source of ice sheet/glacier runoff as ice-sheet-sourced meltwater would decrease surface ocean $\delta^{18}O$ and release ice rafted debris. A proxy that has been used to identify meltwater of continental origin is Ba/Ca measured in planktic foraminiferal shells. Glacial runoff and iceberg melt or increased riverine input would increase the Ba concentrations in the surface ocean[50], which is thought to be recorded in the shells of planktic foraminifera[51]. For example, relative increases in Ba/Ca values in *N. pachyderma* from the central Arctic Ocean have been used to indicate meltwater of a continental origin during the last deglaciation[52]. However, recent studies have shown that Ba/Ca in non-spinose planktic foraminiferal species are much higher than expected, which was related to the foraminiferal calcification microenvironment taking place in marine organic aggregates meaning that shell Ba/Ca ratios are also dependent on primary productivity[53]. This makes a

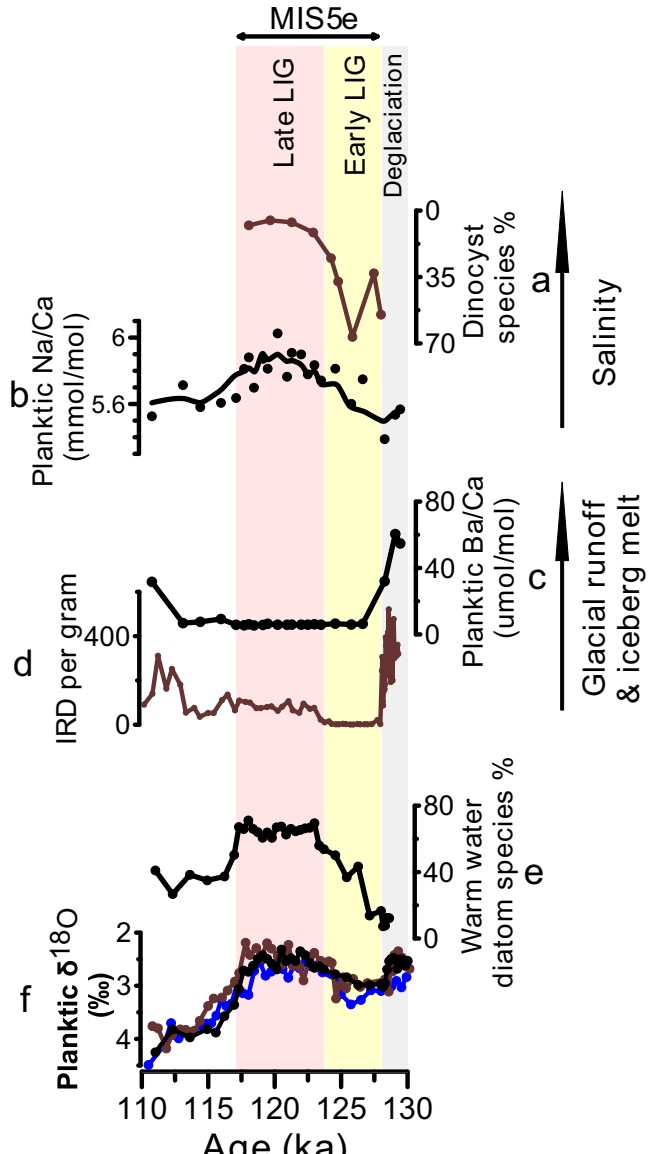

**Fig. 4 | Freshwater anomalies and their potential sources during the Last Interglacial and penultimate deglaciation.** Salinity related proxies (**a**) Relative abundance of the dinocyst species *Brigantedinium* spp. (see Fig. 2g for the relative abundance of other key dinocyst species; ref. 27); and (**b**) Na/Ca measured in the shells of planktic foraminiferal species *Neogloboquadrina pachyderma* (circles represent raw data and the line represents 3-point moving averages). Glacier and ice-berg meltwater proxies: (**c**) Ba/Ca measured in *N. pachyderma;* and (**d**) content of Ice Rafted Debris (IRD) counted in > 150 µm size fraction[23]. **e** Relative abundance of warm-water indicating diatom species[29]. **f** δ18O measured on *N. pachyderma*. Black, blue and brown colors refer to data from sediment cores JM11-FI-19PC, LINK16 and MD95-2009, respectively (see Fig. 1a for core locations). MIS and LIG stand for Marine Isotope Stage and the Last Interglacial, respectively.

quantitative assessment of a potential freshwater anomaly based on foraminiferal Ba/Ca difficult, and we will thus use the non-spinose *N. pachyderma* Ba/Ca as a qualitative indicator of freshwater of continental origin. Our Ba/Ca data measured in *N. pachyderma* show extremely elevated values (30–60 umol/mol) during the penultimate deglaciation (Fig. 3c), likely suggesting influence of glacial runoff from ice sheets. This is also supported by low planktic foraminiferal δ18O and sea surface δ18O (Fig. 4f; ref. 24) as well as high IRD content (Fig. 4d). Across the LIG, planktic Ba/Ca are relatively low and stable (6 µmol/mol, standard deviation = 0.3 µmol/mol, n = 15; Fig. 4c) which does not

support significant influence of meltwater of continental origin (e.g., from ice sheets) during the early LIG compared to the late LIG. A freshwater source that may not result in elevated surface ocean Ba concentration or lower seawater δ18O (i.e., in agreement with our planktic foraminiferal δ18O and Ba/Ca) is sea ice sourced meltwater[54,55]. We therefore suggest that enhanced melting of sea ice in the central Arctic Ocean during the early LIG, likely due to the high solar insolation, was probably the main source of meltwater carried southward with the EGC and transported to the southern Norwegian Sea via the East Icelandic Current (Fig. 1a). This also suggests that this enhanced melting of central Arctic Ocean sea ice during the early LIG resulted in southward advection of buoyancy anomalies potentially reducing open ocean convection and the associated northward ocean heat transport.

One remaining question is if we can provide evidence for a reduction in open ocean convection during the early LIG compared to late LIG and pre-industrial intervals? Benthic foraminiferal assemblages have been suggested to be sensitive to changes in deep water formation rates and/or processes. For example, previous studies from the Norwegian Sea found a close correlation between the composition of benthic foraminiferal assemblages and climate for the past 150 kyr— and identified three main benthic foraminiferal assemblages as "interglacial", "glacial stadial" and "glacial interstadial" fauna[23,56]. The Interglacial assemblage consists of *Cassidulina neoteretis, Cassidulina reniforme, Melonis barleeanus,* and *Islandiella norcrossi.* However, these studies did not examine in detail the potential changes in the benthic fauna within the interglacial periods. Our results show that the Holocene benthic fauna is dominated by *C. neoteretis* and *C. reniforme* constituting together ~60–80% with up to ~20% *M. barleanus* (Fig. 3c). This faunal composition is similar to the late LIG, but the relative abundances of these species for the early LIG are distinctly different. During the early LIG, both *C. neoteretis* and *C. reniforme* constitute < 10% of the benthic foraminiferal assemblages with up to ~50% *M. barleanus* (Fig. 3c). Although, these differences in the foraminiferal fauna are not well understood, they point to different bottom water conditions (likely linked to deep water formation processes and/or rates) compared to the Holocene and late LIG. Changes in productivity and food availability are unlikely explanations for these changes as infaunal benthic δ13C values are the same for both the late and early LIG (see Supplementary Fig. 3). The benthic foraminiferal fauna in the early LIG resemble those of the Younger Dryas stadial (12.8–11.5 ka; ref. 57), when deep water formation in the Nordic Seas is thought to be suppressed[58]. Nevertheless, distinct differences exist between the two time intervals e.g., during the early LIG there was open ocean conditions all-year round, no IRD, and no reported relatively low planktic and benthic δ18O in the southern Norwegian Sea, which is in contrast to the Younger Dryas interval[24,35,57]. Future studies may utilize more proxies such as the water mass mixing proxy, Neodymium isotopes, to further elucidate the differences between reconstructed (de)glacial and interglacial variability in the Nordic Seas deep ocean circulation.

The early LIG represents a case study of a prolonged warmer-than-preindustrial global climate mainly due to stronger northern hemisphere solar insolation. Our findings suggest that southward flux of Arctic buoyancy anomalies (likely related to enhanced melting of sea ice in the central Arctic Ocean) during the Early LIG (~128–124 ka) have altered deep water formation rates and/or processes in the Nordic Seas and thus suppressed northward ocean heat transport (i.e., reduced Nordic Seas heat pump). This highlights the sensitivity of Nordic Seas overturning circulation and regional climate to buoyancy anomalies under interglacial conditions. Satellite observations indicate that the sea ice cover in the Arctic Ocean has declined substantially during the past four decades and it is predicted that consistently ice-free summer conditions will occur by mid-century ~2050 (e.g., refs. 59,60) with serious implications for the climate and ecosystems, though in unpredictable ways (e.g., refs. 3–8). Our study showcases the complex

feedback interactions between a warming climate and Arctic sea ice, and identify the early LIG as a key time interval for data-model inter-comparison efforts to better understand and constrain the impacts of a changing cryosphere on regional and global climate.

## Methods

### Chronology

We used the age models of Capron et al. (ref. 26) for sediment cores MD95-2009 and ENAM33. In brief, the North Atlantic ENAM33 core was transferred onto the AIC2012 ice core chronology[61] by aligning its sea surface temperature to both Greenland ice core $\delta^{18}O$ (as a proxy for Greenland air temperatures) and global abrupt methane increases. This is based on the observation that during the last glacial abrupt climate change, North Atlantic Sea surface temperature increased (semi) simultaneously with both air temperatures over Greenland and atmospheric methane[25,26,62,63]. For the Norwegian Sea MD95-2009 core, Capron et al. (ref. 26) used several lines of evidence to place MD95-2009 on the age model of ENAM33, which we also adopted here to update the age models for the other Norwegian Sea sediment cores LINK16 and JM11-FI-19PC (Supplementary Fig. 4; Supplementary Table 1). Capron et al. (ref. 26) identified the following tie points to reconstruct the age model of MD95-2009: (1) the onset of deglacial decrease in benthic $\delta^{18}O$ records, which is dated at $138.2 \pm 4$ ka; (2) the biostratigraphic link of disappearance of Atlantic benthic foraminiferal species group in both MD95-2009 and ENAM33 marking the onset of the LIG (ref. 23), which is dated at $128 \pm 1.5$ ka in ENAM33 and also matches a remarkable increase in benthic $\delta^{18}O$ in the Norwegian Sea cores (ref. 23, Supplementary Fig. 4b); (3) the ash layer 5e-Low/BasIV identified in both ENAM33 and MD95-2009 (ref. 23) and dated to $123.7 \pm 2$ ka in ENAM33 (ref. 26). This tephra layer is also present in cores LINK16 and JM11-FI-19PC (refs. 24,31); and (4) a pronounced cooling in MD95-2009 SST record is tied to the corresponding enhanced cooling in the NGRIP ice core at $116.7 \pm 2$ ka marking the end of the LIG, which can be transferred to other Norwegian Sea sediment cores by aligning % *N. pachyderma* and planktic $\delta^{18}O$ records (Supplementary Fig. 4a, c). For details see Govin et al. (ref. 25) and Capron et al. (ref. 26). Sediment core JM-FI-19PC does not cover the entire penultimate deglaciation and its bottom part is dated to 130 ka based on the correlation of its planktic and benthic $\delta^{18}O$ records with the MD95-2009 records; this tie point is also transferred to sediment core LINK16 (Supplementary Fig. 4). Age models were then constructed by linear interpolation (i.e., assuming constant sedimentation rate) between tie-points.

### Biomarker analyses

In sediment cores JM-FI-19PC and LINK 16 analyses of brassicasterol (24-methylcholesta-5, 22E-dien-3β-ol), dinosterol (4a-23,24-trimethyl-5a-cholest-22E-en-3β-ol), $IP_{25}$, and $C_{37}$ methyl alkenones were carried out on freeze-dried and homogenized sediments. The samples were extracted with dichlormethane/methanol (2:1, v/v) by ultrasonication ($3 \times 15$ min). For quantification of the lipid compounds, the internal standards 7-HND (7-hexylnonadecane), $C_{36}$ *n*-alkane, and androstanol (5α-androstan-3β-ol) were added prior to any analytical step. The extracts were separated into hydrocarbon and sterol (alkenone) fractions by open silica gel column chromatography using 5 ml *n*-hexane and 9 ml ethylacetate/*n*-hexane, respectively. Furthermore, an aliquot portion of the sterol fraction was derivatized with 200 µl bis-trimethylsilyl-trifluoracet-amid (BSTFA) (60 °C, 2 h).

After extraction with hexane, analyses of sterols and $IP_{25}$ were carried out by gas chromatography-mass spectrometry (GC–MS) using an Agilent 6850 GC (30 m DB-1 MS column, 0.25 mm inner diameter, 0.25 µm film thickness) coupled to an Agilent 5975 C VL mass selective detector. Alkenones were analysed by GC Agilent 6890 A FID equipped with a cold injection system (60 m DB-1MS column, 0.32 mm i.d., 0.25 µm film thickness; cold). In both cases helium was used as carrier gas.

For sterols and $IP_{25}$, individual compound identification was based on comparisons of their retention times with those of reference compounds and on comparisons of their mass spectra with published data[30,64,65]. $IP_{25}$ was quantified using its molecular ion m/z 350 in relation to the abundant fragment ion m/z 266 of 7-HND. Brassicasterol and dinosterol were quantified as trimethylsilyl ethers using the molecular ions m/z 470 and m/z 500, respectively, in relation to the molecular ion m/z 348 of androstanol (for further details see ref. 66).

The $PIP_{25}$ indices were calculated after Müller et al. (ref. 41) using the following equation:

$PIP_{25} = $ conc. $IP_{25}$/[conc. $IP_{25}$ + (conc. phytoplankton biomarker x c)],

where c is a balance factor, calculated by the ratio of mean $IP_{25}$ concentration to mean phytoplankton marker concentration, to counter-balance the higher concentrations of sterols compared to $IP_{25}$. Open water phytoplankton markers brassicasterol and dinosterol were used to calculate $P_BIP_{25}$ and $P_DIP_{25}$ indices, respectively. As both sea ice indices run almost parallel, only $P_BIP_{25}$ is shown.

### Faunal assemblages

In this study we use published and new benthic foraminiferal assemblage records; data from core LINK16 (Fig. 3c, right plot) are new and data from core JM-FI-19PC (Fig. 3c, left plot) are shown in the supplements in Ezat et al. (ref. 67), but not discussed there. When available, at least 300 benthic foraminiferal specimens from the size fraction >100 µm were counted and identified to species level.

Planktic foraminiferal assemblages discussed in this study are published in Abbott et al. (ref. 31) (sediment cores LINK 16), Ezat et al. (ref. 24) (sediment core JM11-FI-19PC), Rasmussen et al. (ref. 23) (sediment cores MD95-2009 and ENAM33). In these studies, at least 300 planktic foraminiferal specimens from the size fraction >100 µm were counted and identified to species level. The relative abundance of planktic foraminiferal species in core JM11-FI-19PC was not quantified for the penultimate deglaciation and LIG parts, but our visual inspection confirms the same patterns as shown from the nearby core MD95-2009. In particular, we did not observe presence of subpolar planktic foraminiferal species in JM11-FI-19PC during the early LIG similar to MD95-2009.

Diatom data discussed in this study are published and described in Hoff et al. (ref. 29).

### Stable Oxygen and Carbon Isotope Analyses

For Core LINK 6, ~25 pristine specimens of the planktic foraminiferal species *Neogloboquadrina pachyderma* and ~15 pristine specimens of the benthic foraminiferal species *Melonis barleeanus* were picked from the size fraction 150−250 um for stable oxygen and carbon isotope analyses. Stable isotope measurements were performed at the Mass Spectrometer Laboratory at UiT the Arctic University of Norway on a Thermo Scientific MAT253 IRMS with a Gasbench II. The precision of the instrument is 0.07‰ for $\delta^{13}C$ and 0.08‰ for $\delta^{18}O$ and results are reported on the VPDB standard. Planktic and benthic $\delta^{18}O$ data from JM11-FI-19PC core are published in Ezat et al. (ref. 24) and MD95-2009 data are published in Rasmussen et al. (ref. 23).

### Planktic foraminiferal Na/Ca and Ba/Ca analyses

Only pristine 60−160 specimens of the planktic foraminiferal species *N. pachyderma* with no visible signs of dissolution were selected from the size fraction 150−250 µm for minor/trace element analyses from sediment core JM11-FI-19PC. The specimens were gently crushed, weighed, and cleaned following the "full cleaning" procedure of Martin and Lea (ref. 68). This included the following steps: 1) Milli-Q water and methanol rinses to remove clay contaminates; 2) oxidation of organic matter by buffered $H_2O_2$; 3) reduction step with buffered solution of anhydrous hydrazine to remove Mn-Fe oxide coatings; 4) treatment with alkaline diethylene-triamine-pentaacetic acid (DTPA) to remove sedimentary barium contaminants e.g., barite; and 5) brief weak acid

leach. The cleaned foraminiferal samples were then dissolved and analyzed on iCAPQ Inductively Coupled Plasma Mass Spectrometry (ICP-MS) at Lamont Doherty Earth Observatory (LDEO) at Columbia University, USA. Based on repeated measurements of in-house standard solutions, the average relative precision is 0.8% and 2.1% for Ba/Ca and Na/Ca, respectively. Blank samples were analyzed within every batch of samples in order to monitor potential contamination from reagents and vials. The Mg/Ca and B/Ca data were previously published in Ezat et al. (ref. 24) and the Ba/Ca and Na/Ca data are new to this study. As discussed in Ezat et al. (ref. 24), the Mg/Ca based temperatures also indicate early LIG cooling in line with the faunal planktic foraminiferal and diatom assemblages.

## Data availability
The authors declare that all original data presented in this article are available at UiT Open Research Data Repository (https://dataverse.no/dataset.xhtml?persistentId=doi:10.18710/3LCQJX) and within the article Supplementary files (Supplementary Dataset 1).

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

## Acknowledgements

We thank M. Lindgren, N. El bani Altuna, T. Dahl, I. Hald, K. Monsen, B. Honish, J. Ruprecht, L. Pena, K. Esswein, and W. Luttmer for laboratory support. We also thank E. Capron for sharing data from sediment core MD95-2009. This study is financed by a starting grant from the Tromsø Forskningsstiftelse to M.M.E., project number A31720. The research also received support from the Research Council of Norway and the Co-funding of Regional, National, and International Programmes (COFUND)–Marie Skłodowska-Curie Actions under the EU Seventh Framework Programme (FP7), project number 274429. M.M.E. is also part of the Centre of Excellence iC3, grant number 332635 and the ERC synergy project i2B, grant number 101118519.

## Author contributions

M.M.E. conceived and designed the study, performed most of the foraminiferal geochemical analyses and wrote the manuscript. T.L.R. provided the isotope data and benthic foraminiferal assemblages from core

LINK16. K.F. conducted the biomarker analyses and their evaluation and quality control. M.M.E., T.L.R., and K.F. contributed to the discussions and the final draft of the manuscript.

## Funding

## Competing interests
The authors declare no competing interests.
