## [Transparent Peer Review file · Nature Communications]

Arctic freshwater outflow suppressed Nordic Seas overturning and oceanic heat transport during the Last Interglacial

Corresponding Author: Dr Mohamed Ezat

Version 0:

Reviewer comments:

Reviewer #1

(Remarks to the Author)

Review:

Last Interglacial Arctic Sea ice, ocean circulation and regional climate

Authors: Mohamed M. Ezat^{1*}, Kirsten Fahl², Tine L. Rasmussen

General. This paper addresses the complexity of the last interglacial period in the Arctic Ocean and adjacent Nordic Sea with particular emphasis on AMOC circulation. It is an important topic for understanding modern Arctic warming and AMOC slowdown and the paper is well written and the study carefully planned and carried out.

Points for authors to consider:

Most important, the title says Arctic sea ice and refers to the central Arctic early in the paper. But the studied cores are from Norwegian Sea/Iceland area. While we recognize the close link for inflowing Atlantic Layer water, outflowing cold E Greenland water, etc, between Arctic Ocean proper and Nordic Seas [Greenland, Iceland, Norwegian Seas], Perhaps clarity for those readers unfamiliar with this complex system would benefit from definitions early in the paper. This is subtle, Arctic Ocean sea ice flowing out into the Nordic Seas, should be clarified.

I also noticed that many studies of the LIG in the central and western Arctic are not cited. Recognizing, central arctic sedimentation is slower and intermittent, there are nonetheless records. Check deVernal, Vermassen, Polyak Cronin and others on interglacial conditions in the central Arctic [including a summer ice free LIG]. This also raises the question: if the authors make direct comparisons between LIG and Holocene interglacial, perhaps the large Holocene literature for Greenland and the Arctic should be covered in more detail.

The discussion of benthic foram assemblage variability within the interglacial is important as McManus pointed out many years ago for the N Atlantic. Do the authors view MIS5 variability – prominent in the three LIG phases they define - to be related to sea or land ice and Heinrich type millennial behaviour? The similarity of LIG to the YD seems important to me.

In summary, I think this statement in discussion section justifies publication of this study:

“This highlights the sensitivity of Nordic Seas open ocean convection and regional climate to buoyancy anomalies under interglacial conditions.”

Minor points

LIG is defined as 129-117 ka early then 128–117 ka in figure caption 3.

Reviewer #2

(Remarks to the Author)

Last Interglacial Arctic Sea ice, ocean circulation and regional climate

This manuscript argues that freshwater outflow from the Arctic, derived from melting sea ice, delayed warming during the last interglacial (LIG) by slowing deep-water formation in the Nordic seas.

It presents foram assemblage and isotope data, trace metal chemistry and biomarker-based sea ice reconstructions from three sites in the southern Nordic Seas, adjacent to Iceland. The late onset of warming during the LIG is a well-documented occurrence in Nordic Sea records – but I believe that the authors suggestion that it is caused by outflow of sea-ice derived meltwater is a unique argument.

Overall, I think it is an intriguing hypothesis and easy-to-read manuscript, but there are a number of shortcomings that I think the authors could easily correct in a revised submission. I do think the topic, and manuscript would have broad appeal and be a significant contribution to Arctic and sub-Arctic paleoceanography.

1. The map embedded in figure 1 lacks a bit of detail. I think the authors need to better elevate the past work that has been done on this topic, and show some of the key records from the Nordic seas where this division between early and late LIG proxy data exists. This is especially true for some of the records mentioned in the manuscript but have no geographic information attached to them. For example, Interpreting a SST record from the Nordic seas requires one to know which water mass it was likely sitting in (Atlantic, Arctic?)

2. I think it is necessary to show (at least in the supplementary information) the sterols concentrations in the cores. Currently we are only shown the IP25 ug/g and the PBIB25 index. It is not possible to see if there is any biomarker-based evidence for open water productivity during the LIG – which is kind of important for deciding whether the IP25 records for sea ice are reliable or not. In a way this is buried in the PBIP25 index, but it would also make sense to show the raw sterol concentrations somewhere.

3. On lines 198-200 the authors discuss interpreting the %C37:4 and Na/Ca concentrations of foram shells as proxies for sea surface salinity. This seems to make sense, but can they also comment on how this fits with existing dinocyst-based SST reconstructions from the Nordic Seas?

4. Another 'proxy' that needs a better introduction/explanation is Ba. On Line 70 the authors write: "Glacial runoff and iceberg melt would increase the Ba concentrations in the surface ocean³⁸." This is a key piece of the argument, and as such I would like to see a more substantial statement here, that perhaps includes some typical concentrations compared to open ocean sites, but also some examples of where this has been applied. Ultimately, it is the Ba data that seems to be used to rule out other glacial sources for meltwater or river outflow etc. Can the authors develop these arguments in a more convincing and rigorous way? Ultimately, at the end of the manuscript, I am not clear on why it must be Arctic sea ice melt that added freshwater to the Nordic seas to suppress deep-water convection. The reason is the lack of detail in how some of the results and proxies are presented. They lack of bit of rigor in their explanations that could easily be fixed.

6. Finally, I wonder about the concluding sentiment here – that persistent melting of Arctic sea ice during the the early LIG was a significant source of meltwater to the Nordic Seas. Would it be possible to quantify (back of the envelope) how much sea ice melt would be needed? The reason I ask is that today, and probably during most Quaternary interglacials, there has been persistent melting of winter sea ice in the Arctic. So conceptually, how does this differ from what the authors envisage during the LIG? This is a critical question that needs to be resolved before a final decision can really be made on the impact and viability of this manuscript. Currently, it is a rather qualitative description – and I think it needs to be a bit more.

Overall, I am very positive about this contribution and feel the authors should be given a chance to re-submit after a moderate revision. I have attached an annotated PDF with some additional comments and edits.

Version 1:

Reviewer comments:

Reviewer #1

(Remarks to the Author)

I read the response to both reviewers and the revised manuscript - I find the new draft acceptable and recommend NatComm publish this.

Response letter

The authors wish to thank the two reviewers for their very constructive comments. Below we explain how we implemented each comment in the manuscript; *our responses are in bold and italic font.*

REVIEWER COMMENTS

Reviewer #1 (Remarks to the Author):

Review:

Last Interglacial Arctic Sea ice, ocean circulation and regional climate

Authors: Mohamed M. Ezat^{1*}, Kirsten Fahl², Tine L. Rasmussen

General. This paper addresses the complexity of the last interglacial period in the Arctic Ocean and adjacent Nordic Sea with particular emphasis on AMOC circulation. It is an important topic for understanding modern Arctic warming and AMOC slowdown and the paper is well written and the study carefully planned and carried out.

Points for authors to consider:

Most important, the title says Arctic sea ice and refers to the central Arctic early in the paper. But the studied cores are from Norwegian Sea/Iceland area. While we recognize the close link for inflowing Atlantic Layer water, outflowing cold E Greenland water, etc, between Arctic Ocean proper and Nordic Seas [Greenland, Iceland, Norwegian Seas], Perhaps clarity for those readers unfamiliar with this complex system would benefit from definitions early in the paper. This is subtle, Arctic Ocean sea ice flowing out into the Nordic Seas, should be clarified.

We changed the title accordingly to “Arctic freshwater outflow suppressed Nordic Seas overturning and oceanic heat transport during the Last Interglacial”. Also, we now defined what we mean by central Arctic Ocean, Nordic Seas earlier in the ‘introduction section’, and also the reference to different areas clearer throughout the manuscript (see e.g., lines 44, 60, 20).

I also noticed that many studies of the LIG in the central and western Arctic are not cited. Recognizing, central arctic sedimentation is slower and intermittent, there are nonetheless records. Check deVernal, Vermassen, Polyak Cronin and others on interglacial conditions in the central Arctic [including a summer ice free LIG]. This also raises the question: if the authors make direct comparisons between LIG and Holocene interglacial, perhaps the large Holocene literature for Greenland and the Arctic should be covered in more detail.

We are now referring and discussing more studies from the central Arctic Ocean, including Polyak et al., 2013; Nørgaard-Pedersen et al., 2007; Adler et al., 2009, Kageyama et al., 2021, Stein et al., 2017; Vermassen et al., 2023; Hillaire-Marcel et al. 2017; Razmjooei et al., 2023; de Vernal et al., 1994 (see e.g., line 58-67). Regarding the comparison to the Holocene, our study is about the LIG development in the region but the comparison to the Holocene is only to show that that the early LIG was colder (and has different benthic foraminiferal assemblages) compared to the late LIG and

the Holocene at the studied area. We have now changed the title of the caption of figure 3 to make this point clearer.

The discussion of benthic foram assemblage variability within the interglacial is important as McManus pointed out many years ago for the N Atlantic. Do the authors view MIS5 variability – prominent in the three LIG phases they define - to be related to sea or land ice and Heinrich type millennial behaviour? The similarity of LIG to the YD seems important to me.

We thanks the reviewer for this comment which enabled us to provide more insightful information. The glacial stadials (including Heinrich stadials) are characterized by distinctly different benthic foraminiferal assemblages compared to the early and late LIG – which is described in details in the references we refereed in this part (Rasmussen et al., 1996, 1999, 2003). Despite the similarity in the benthic assemblages between the deglacial YD event and the early LIG, clear differences exist between the two time intervals which we have now highlighted in the manuscript: “Nevertheless, distinct differences exist between the two time intervals e.g., during the early LIG there was open ocean conditions all-year round, no IRD, and no reported relatively low planktic and benthic $\delta^{18}O$ in the southern Norwegian Sea, which is in contrary to the Younger Dryas interval^{57,24,35}. Future studies may utilize more proxies such as the water mass mixing proxy, Neodymium isotopes, to further elucidate the differences between reconstructed (de)glacial and interglacial variability in the Nordic Seas deep ocean circulation (see lines 265-270)”.

In summary, I think this statement in discussion section justifies publication of this study: “This highlights the sensitivity of Nordic Seas open ocean convection and regional climate to buoyancy anomalies under interglacial conditions.”

Minor points

LIG is defined as 129-117 ka early then 128–117 ka in figure caption 3.

We have now made it consistent throughout referring to the LIG as ~128-117 ka – as the onset of LIG in the Nordic Seas is defined at 128 ± 1.5 ka in Capron et al 2014 – which we also utilized in our age model (see Methods).

Reviewer #2 (Remarks to the Author):

Last Interglacial Arctic Sea ice, ocean circulation and regional climate

This manuscript argues that freshwater outflow from the Arctic, derived from melting sea ice, delayed warming during the last interglacial (LIG) by slowing deep-water formation in the Nordic seas. It presents foram assemblage and isotope data, trace metal chemistry and biomarker-based sea ice reconstructions from three sites in the southern Nordic Seas, adjacent to Iceland. The late onset of warming during the LIG is a well-documented occurrence in Nordic Sea records – but I believe that the authors suggestion that it is caused by outflow of sea-ice derived meltwater is a unique argument. Overall, I think it is an intriguing hypothesis and easy-to-read manuscript, but there are a number of shortcomings that I think the authors could easily correct in a revised submission. I do think the topic, and manuscript would have broad appeal and be a significant contribution to Arctic and sub-Arctic paleoceanography.

1. The map embedded in figure 1 lacks a bit of detail. I think the authors need to better elevate the past work that has been done on this topic, and show some of the key records from the Nordic seas where this division between early and late LIG proxy data exists. This is especially true for some of the records mentioned in the manuscript but have no geographic information attached to them. For example, Interpreting a SST record from the Nordic seas requires one to know which water mass it was likely sitting in (Atlantic, Arctic?)

We have now described the geographic location of these records in the manuscript (see lines 76-77, 83-88) and referred to the map. Also, the reviewer kindly made relevant comments in an annotated file, please see our responses to these respective comments below.

2. I think it is necessary to show (at least in the supplementary information) the sterols concentrations in the cores. Currently we are only shown the IP25 ug/g and the PBIB25 index. It is not possible to see if there is any biomarker-based evidence for open water productivity during the LIG – which is kind of important for deciding whether the IP25 records for sea ice are reliable or not. In a way this is buried in the PBIP25 index, but it would also make sense to show the raw sterol concentrations somewhere.

We are now showing the sterol concentrations and other sea ice indicators in the Supplemental Figure 2. We also included the sterol concentrations in the discussion (see lines 158-173).

3. On lines 198-200 the authors discuss interpreting the %C37:4 and Na/Ca concentrations of foram shells as proxies for sea surface salinity. This seems to make sense, but can they also comment on how this fits with existing dinocyst-based SST reconstructions from the Nordic Seas?

We followed the reviewer's suggestion and now added the available dinocyst data from our studied area (e.g., see Figure 2g), which indeed agree with other data/proxies discussed in our manuscript. Now we are discussing the dinocysts within the 3 different sections of 'temperature', 'sea ice' and 'salinity' discussions (e.g., see lines 113-116, 170-171, 204-208).

4. Another 'proxy' that needs a better introduction/explanation is Ba. On Line 70 the authors write:

“Glacial runoff and iceberg melt would increase the Ba concentrations in the surface ocean³⁸.” This is a key piece of the argument, and as such I would like to see a more substantial statement here, that perhaps includes some typical concentrations compared to open ocean sites, but also some examples of where this has been applied. Ultimately, it is the Ba data that seems to be used to rule out other glacial sources for meltwater or river outflow etc. Can the authors develop these arguments in a more convincing and rigorous way? Ultimately, at the end of the manuscript, I am not clear on why it must be Arctic sea ice melt that added freshwater to the Nordic seas to suppress deep-water convection. The reason is the lack of detail in how some of the results and proxies are presented. They lack of bit of rigor in their explanations that could easily be fixed.

We have now added more information about foraminiferal Ba/Ca as a proxy for freshwater of continental origin (see lines 218-228). Also, in response to this comment, we have now significantly restructured the discussion part of early LIG freshening and its sources to make our arguments clearer. We are now first discussing salinity changes across LIG using $\delta^{13}C_{37:4}$ and Na/Ca as well as published dinocyst data from the area. Then we discuss potential sources for early LIG freshening by listing different sources and how different proxies are in apparent disagreement with all potential sources except for sea ice melt (i.e., enhanced melting of central Arctic sea ice and southward export via EGC and EIC). Finally, we discuss the evidence for a changing deep water formation/condition across the LIG.

6. Finally, I wonder about the concluding sentiment here – that persistent melting of Arctic sea ice during the the early LIG was a significant source of meltwater to the Nordic Seas. Would it be possible to quantify (back of the envelope) how much sea ice melt would be needed? The reason I ask is that today, and probably during most Quaternary interglacials, there has been persistent melting of winter sea ice in the Arctic. So conceptually, how does this differ from what the authors envisage during the LIG? This is a critical question that needs to be resolved before a final decision can really be made on the impact and viability of this manuscript. Currently, it is a rather qualitative description – and I think it needs to be a bit more.

We have now replaced the word ‘persistent’ with ‘enhanced’, please see also our response to the respective comment in the annotated PDF below. We have now added more text to this part further highlighting the fact that stronger northern hemisphere solar insolation during the early LIG than during the Holocene. We also agree that our hypothesis should be more tested by e.g., numerical modelling, and we are currently in a contact with earth system modelling group to take it further; we refer to this in the manuscript via “Our study showcases the complex feedback interactions between a warming climate and Arctic sea ice and identify the early LIG as a key time interval for data-model intercomparison efforts to better understand and constrain the impacts of a changing cryosphere on regional and global climate”.

Overall, I am very positive about this contribution and feel the authors should be given a chance to re-submit after a moderate revision. I have attached an annotated PDF with some additional comments and edits.

We transferred the additional comments from the annotated pdf with their respective line numbers there to this file – please, see below. As above, in our answer, the line numbers refer to lines in the revised manuscript.

Line 20. I am not aware of issues with correlative records. Dating is certainly an controversial issue - but perhaps more acutely, the kind of resolution one needs to look at a few thousand year offsets in climate phenomena simply have not been recovered from the central Arctic Ocean, and in the absence of a continuous isotope stratigraphy would be difficult to date at such a resolution. The use of 'correlation' here does not seem to capture these aspects of the problem.

At the same time (and perhaps this is discussed later) - even in low sedimentation rate settings, where presumably most of the terrigenous material on the seafloor is deposited from sea ice transport, open water conditions in the central Arctic even for a small portion of the LIG would/could leave a discernible signal in either biomarker concentrations or microfossil populations. For example, in the appearance in subpolar specialists in different taxa that would indicate a substantially different environment than exists in any modern or Holocene analogues.

Therefore, it is not sufficient to dismiss Arctic marine sediment archives because of these perceived limitations. Any inference of sea-ice free conditions in the Arctic (based on sub-Arctic records) needs to be anchored by a discussion of the available proxy data and interpretations that exist from the Arctic. I have not read beyond the abstract yet, so I mention this now and will see how this aspect is dealt with by the authors.

We totally agree that the central Arctic records provide invaluable insights. We didn't edit the sentence as it is now deleted in order to meet the maximum word limit (150 words) for the abstract

Line 27. Does this mean that northward heat transport was also suppressed.

Yes – we slightly edited the sentence to make it clearer (lines 28-30).

Line 42: Here it would be good to define what is moving south. Positive buoyancy anomalies is not so descriptive - is it less saline and warmer surface water (causing positive buoyancy anomalies???)

Done (line 45).

Line 54. This sentence could be far more descriptive and informative by saying like . . . "organic geochemical proxies suggest the existence of a perennial sea ice cover during the LIG (Stein et al XXX), while the occurrence of sub-polar planktic foraminifera in sediments assigned to the LIG from across the central Arctic have supported arguments for sea-ice free conditions (Vermassen et al., 2023)" Otherwise it paints a rather pessimistic picture by not including relevant information.

Done (see lines 61-64)

Line 56. This should be re-written - it comes off very awkward. The central point is true, that nobody has tried to dissect LIG records from the central Arctic into 1-2 kyr timeslices because either a) the sedimentation rates are too low (less than 1-2 cm/kyr) or b) the recognition of a more pressing problem in that the identification of stratigraphic position for the LIG remains debated as different dating tools continue to provide different age assignments to central Arctic sediments (as discussed in the following sentence).

Done (lines 67-70).

Line 63. These sentences are really important for framing the past work but they lack detail - for example the location of the studied cores, the conditions that they are exposed to today etc. furthermore - although the difficulty of reconstructing sea ice in the central Arctic has been mentioned, there is no information on what the current status of LIG sea ice reconstructions are in the Nordic Seas. To me this seems very surprising given the scope of the paper and could presumably be easily fixed.

We have now added this information later in this paragraph (see the second last paragraph in the "Introduction" section)

Line 64. it seems really important to provide some additional information here. As the location of this SST record is rather critical for interpreting it as reflecting a change in the sea-ice conditions, and why. Was it a western (influenced by EGC) or eastern site (influence by Atlantic water inflow) in the Nordic seas? I think this information can be worked into this sentence easy enough, and perhaps this site can be named and shown on figure 1?

Done (see lines 76-77).

Line 66. Is this the first time that this argument has been put forward to explain the SST data at this site? Or is there a reference for this?

We are not aware of any study that explicitly mentioned the possibility of presence of winter sea ice in the southern Norwegian during the early LIG

Line 70. Yes, but where?

We now added the core name and referred to its location on the map (see lines 83-84).

Line 143. It would be very nice to see the concentration data Ip25 and the open-water sterols commented on in the text - or shown in the main paper (not the supp. information) - as a lot of the sea ice arguments hinges on 'indices' that combine these datasets - and when the indices suggest values of '0' it is nice to see why

We have now described sterol concentrations in the text and showed it in a Supplemental Figure with more sea ice indicators. Please see also our response comment #2 from Reviewer #2 above.

Line 152. What is the evidence for this? This sentence loses me a little . . .

The sentence in question is now deleted (175-179).

Line 157. This sounds really important - but it is hard to gauge the novelty here as the authors have not tried to summarize the state-of-the-art for sea ice conditions in the Nordic seas during the LIG. This really seems like a large omission that makes it hard to evaluate the manuscript.

In also a response to a previous comment, we have now described in the introduction the status of Nordic Seas sea ice reconstructions during MIS 5 (see our responses above to Reviewer#2 comments on lines 63, 66).

Line 175. Usually this paper is discussing the early or late LIG. It is confusing when the entire 'LIG' is referenced. Perhaps, in such a case, instead of 'During the LIG' it should be 'Across the LIG' - but I am confused none-the-less whether this sentence refers to the early LIG or the entire LIG.

We mean the entire LIG and we followed the suggestion and replaced “During” with “Across”. We have also edited this part of the argument to make it clearer (lines 232-234). Also, see our response to comment #4 of Reviewer #2 above.

Line 184. This sentence seems to contradict the next. Can the authors better explain why they think the $\delta^{18}O$ and Ba data from surface waters around Iceland likely indicate the melting of sea ice in the central Arctic ocean? It's just not clearly there in the two sentences leading up to this pivotal sentence. I also think it is important to more explicitly define these proxies - like Ba - with a sentence that describes what it is a proxy for and why.

Please see our response to comment #4 of reviewer #2. We have now added more information about foraminiferal Ba/Ca as a proxy for freshwater of continental origin (see lines 218-228). Also, we have now significantly restructured the discussion part of early LIG freshening and its sources (Lines 189-270) to make our arguments clearer. We are now first discussing salinity changes across LIG using $\delta^{13}C_{37:4}$ and Na/Ca as well as published dinocyst data from the area. Then we discuss potential sources for early LIG freshening by listing different sources and how different proxies are in apparent disagreement with all potential sources except for sea ice melt (i.e., enhanced melting of central Arctic sea ice and southward export via EGC and EIC). As there is almost no oxygen isotope fractionation during sea ice formation, melting of sea ice will result in salinity decrease but almost no change in seawater $\delta^{18}O$ – and the meltwater source is not of a continental origin i.e. not expected to result in elevated seawater Ba (lines 238-240). Finally, we discuss the evidence for a changing deep water formation/condition across the LIG.

Line 188. When you say persistent - what do you mean? Sea ice melts in the summer rather persistently. So what is envisioned during the early LIG? Anything beyond what happens today - or does 'persistent' imply something extraordinary? Is the 'extraordinary' aspect that sea ice was present in the Arctic during the early LIG? If so, I am not sure anyone has ever argued it was not there through the winter, and it would have always melted back to some extreme in the summer.

We thank the reviewer very much for this comment. We meant “enhanced” and not “persistent”. We replaced the word “persistent” by “enhanced” (e.g., lines 240, 248)

Line 197. What do the existing dinocyst records from the Nordic sea say about spatial and temporal patterns in sea surface salinity during the LIG?

We followed the reviewer's suggestion and now added the available dinocyst data from our studied area (e.g., see Figure 2g), which indeed agree with other data/proxies discussed in our manuscript. Now we are discussing the dinocysts within the 3 different sections of 'temperature', 'sea ice' and 'salinity' discussions (e.g., see lines 113-116, 170-171, 204-208).

Line 217. These are well articulated arguments for a change in deep-water circulation, and the comparison with the YD is intriguing - it would be fantastic to see this mapped out in a more systematic way in another work.

Thank you - we agree.

Response letter

REVIEWERS' COMMENTS

Reviewer #1 (Remarks to the Author):

I read the response to both reviewers and the revised manuscript - I find the new draft acceptable and recommend NatComm publish this.

Response: thank you for taking the time to read our responses and revised manuscript.

Last Interglacial Arctic Sea ice, ocean circulation and regional climate

**Authors:** Mohamed M. Ezat^{1*}, Kirsten Fahl², Tine L. Rasmussen³

¹ iC3 - Centre for ice, Cryosphere, Carbon and Climate, Department of Geosciences, UiT, The
Arctic University of Norway, Norway.

² Alfred Wegener Institute Helmholtz Centre for Polar and Marine Research, Am
Handelshafen 12, 27570 Bremerhaven, Germany.

³ Department of Geosciences, UiT, The Arctic University of Norway, Norway.

* Correspondence to: M. M. Ezat (mohamed.ezat@uit.no).

12 13 **Abstract**

The Last Interglacial period (LIG), ~129,000–117,000 years ago, was characterized by a long-term
Arctic atmospheric warming above the preindustrial level. The LIG thus provides a case study of
Arctic feedback mechanisms of the cryosphere-ocean circulation-climate system under warm climatic
conditions. Previous studies suggested a delay in the LIG peak warming in the North Atlantic
compared to the Southern Ocean, and evoked the possibility of significant southward extension of sea
ice during the early LIG. Reconstructions of environmental changes in the Arctic Ocean are hampered
by large uncertainties in timing **and correlation** of central Arctic Ocean records, low sedimentation
rates, and method limitations. Here we compile new and published proxy data on past changes in sea-
ice distribution, sea surface temperature and salinity, deep ocean convection, and meltwater sources
based on well-dated records from the Nordic Seas, northern North Atlantic. We identify a distinct
development in sea surface temperature with a cold early LIG followed with a transitional warming
phase and then a warmer-than-present late LIG. Open ocean conditions in the Norwegian Sea
prevailed throughout the LIG all year round. Further, our data suggest persistent melting of sea ice in
the central Arctic Ocean during the early LIG potentially **suppressed deep-water formation and**
**northward heat transport**. Our findings showcase the complex feedback interactions between a
warming climate, sea ice, ocean circulation and regional climate.

31 **Main**

The Arctic cryosphere is transforming rapidly in response to ongoing climate change with
profound implications for regional and global climate, future sea level rise and stability of ecosystems^{1,2}.
For example, it has been suggested that positive heat and freshwater flux anomalies in the Arctic are the
primary cause of the observed recent slow-down of the Atlantic Meridional Overturning Circulation
(AMOC), a crucial regulator of the earth's climate and fundamental for the mild climate of northwest
Europe³. This has also been linked to what is called the subpolar North Atlantic "Warming Hole" where
the ongoing warming is slower than elsewhere on the globe or has even cooled down over recent years³.
A conceptual explanation of these effects is that the associated northward oceanic meridional transport
of heat and salt within the AMOC is balanced by a southward flow of cold deep water that is mainly
formed in the Nordic Seas and the subpolar gyre. Thus, a southward advection of Arctic **positive**
**buoyancy anomalies** may have suppressed the deep-water formation in these key areas thus altering the
AMOC and the associated northward heat transport. However, there is no agreement regarding the
relative role of individual components of Arctic climate change (e.g., melting of the Greenland Ice Sheet,
sea ice reduction-induced freshwater and heat flux anomalies) to the observed slowdown of the AMOC
(refs. 3–5) or the associated changes in deep ocean convection^{6–8}. A broader paleoclimatic perspective,

permitting a wider range of boundary conditions and different rates of climate change to be investigated,
is a key for understanding the interplay and feedback mechanisms between changes in the Arctic
cryosphere, the global ocean- and atmosphere circulation, and climate.

The Last Interglacial period (LIG; ~129–117 ka, thousands of years before present) was
characterized by a warmer-than-present global climate, a smaller ice volume and a higher sea level⁹. It
also provides a case study of long-term polar atmospheric warming above the preindustrial level^{10,11}
and the response of the Arctic climate system and strength of the deep ocean convection to this
warming. However, proxy studies gave conflicting results on changes in Arctic Ocean sea-ice cover
during the LIG indicating either perennial sea ice all year round or fully open ocean conditions during
summers^{12,13}. In addition, the time scales for these Central Arctic marine proxy records are associated
with too large uncertainties to allow correlations of the reconstructions of the development of sea ice
changes e.g., due to varying insolation forcing across the LIG. Further, recent studies have questioned
the LIG age of these records from the central Arctic Ocean^{14,15}. Studies from the central and northern
Nordic Seas have focused on identifying the LIG peak warming and comparison with the Holocene
climate state or variability^{16–18}. Higher resolution and well-dated records from the southern Norwegian
Sea and the subpolar North Atlantic suggested a delay in the LIG peak warming in the North Atlantic
compared to the Southern Ocean^{19–22}. Further, planktic foraminiferal assemblages from the southern
Norwegian Sea suggest summer temperature below 4°C (for example, ref. 22), which raises the
possibility that winter sea ice may have expanded much further south in the Nordic Seas during a
globally warmer-than-present time interval compared to the historical and preindustrial periods.
Although the early LIG cooling was attributed to ice sheet melting and suppression of Nordic Seas
deep convection^{21,23}, neither the sources of meltwater nor deep water formation processes have been
constrained by proxy data. Nevertheless, a more recent study based on diatom assemblages suggested
that the sea surface during the early LIG was warm²⁴, but a detailed comparison between the two sea
surface temperature proxies (i.e., diatom and planktic foraminifera) is still lacking.

[revised manuscript text omitted]

IP_{25} is absent and sea-ice index values are zero indicating open water conditions all year around. The
sea ice development from penultimate deglaciation and across the LIG is also in agreement with sea
ice-related diatom species (ref. 24; Figure 3). Although temperature proxies (diatoms and
foraminiferal assemblages) suggest similar summer sea surface temperatures during the latest part of
penultimate deglaciation and earliest LIG (Figure 2 d,e,f), **winter sea surface temperature must have**
**significantly increased during the sea ice free Earliest LIG compared to penultimate deglaciation**
**winters. This probably suggests diminished seasonality during the early LIG at 128–126.5 ka.** After
the end of the LIG, biomarker (IP_{25} , $P_{BIP_{25}}$) data and sea ice-indicating diatom species suggest
different timing of the appearance of sea ice. The abundance of sea ice indicating diatom species starts
to increase (close to the Termination II levels) immediately at the end of LIG at ~116.5 ka (Figure 2b),
whereas IP_{25} was absent until ~ 114 ka (Fig 2c, d). It could be that winter sea ice started to extend
southward at ~116.5 ka, but it only reached the Faroe Islands margin at 114 ka or later – such a
prolonged open ocean corridor during the last glacial inception may have provided moisture for ice
sheet growth^{35,36,20}.

Constraints on the sources of potential freshwater anomalies and implications

What caused the regional cooling during the early LIG? The early LIG interval (~128–124 ka)
was characterized by high summer solar insolation at the high northern latitudes – significantly higher
than pre-industrial levels (ref. 37; Figure 1f). Previous studies suggested the potential cooling of the
early LIG may have been caused by persistent melting of ice sheets^{21,23} that could have disturbed open
ocean convection in the Nordic Seas and the subpolar North Atlantic and thus suppressed the northward
heat transport. Yet, proxy evidence for an early LIG meltwater event and its source is lacking. Glacial
runoff and iceberg melt would increase the Ba concentrations in the surface ocean³⁸. Ba/Ca measured in
planktic foraminifera thus records past changes in seawater Ba concentration³⁹. Our Ba/Ca data
measured in *N. pachyderma* show elevated values (30–60 $\mu\text{mol/mol}$) during the penultimate deglaciation
(Figure 3c), suggesting influence of glacial runoff from ice sheets. This is also supported by low planktic
foraminiferal $\delta^{18}\text{O}$ and sea surface $\delta^{18}\text{O}$ (Figure 4f; ref. 20) as well as high content of Ice Rafted Debris
(IRD) (Figure 4d). **During the LIG,** planktic Ba/Ca are low and stable (6 $\mu\text{mol/mol}$, standard deviation
= 0.3 $\mu\text{mol/mol}$, n=15) which does not support significant influence of meltwater from ice sheets
compared to the late LIG at ~126.5–124); note that the 128–126.5 ka time interval is not resolved by
our Ba/Ca (Figure 4c). This is also supported by slightly higher planktic $\delta^{18}\text{O}$ and seawater $\delta^{18}\text{O}$ in the
southern Norwegian Sea during the early LIG compared to the late LIG (Figure 4f; ref. 20); ice-sheet-
sourced meltwater would decrease surface ocean $\delta^{18}\text{O}$. Also, there is complete absence of IRD during
the whole of the early LIG (~128–123.5 ka) in the studied cores.

Other sources for freshwater anomalies include local precipitation, meltwater from sea-ice and
changes in the inflow of Atlantic surface water. An increase in local precipitation or a decrease in salt
advection from the south are likely to decrease seawater $\delta^{18}\text{O}$. **Sea-ice sourced meltwater however**
**may not result in elevated surface ocean Ba concentration or lower seawater $\delta^{18}\text{O}$ (refs. 40, 41).** We
therefore suggest that persistent melting of sea ice in the central Arctic Ocean during the early LIG,
likely due to high solar insolation, was probably the main source of meltwater and transported to the
southern Norwegian Sea via the East Icelandic Current (Figure 1a). **This also suggests that persistent**
melting of central Arctic Ocean sea ice during the early LIG, resulted in southward advection of
buoyancy anomalies during the early LIG potentially reducing open ocean convection and the
associated northward ocean heat transport. Nevertheless, evidence for the freshwater water anomaly
(e.g., lower salinity) itself during the early LIG compared to the late LIG remains elusive. To date, it
has proven difficult to develop a reliable proxy for sea surface salinity. The Na/Ca in planktic
foraminifera is proposed as a proxy for surface ocean salinity, although non-salinity parameters may
play a role⁴². Also, there is a strong correlation between relative abundance of the C37 tetra-
unsaturated methyl alkenone (% $C_{37:4}$) and both sea ice and sea surface salinity⁴³, but individual effects
of sea ice and low salinity on % $C_{37:4}$ are not separated. Given that **biomarker and diatom assemblage**
**data suggest no sea ice in the area during the LIG, we suggest that % $C_{37:4}$ is mostly recording changes**

in sea surface salinity for the LIG part. Using %C_{37:4} and Na/Ca measured in shells of *N. pachyderma*
as qualitative indicators for sea surface salinity, both independent proxies suggest lower salinity during
the early LIG compared to the late LIG (Figure 4a,b).

One remaining question is if we can provide evidence for a reduction in open ocean convection
during the early LIG compared to late LIG and pre-industrial intervals? Benthic foraminiferal
assemblages have been suggested to be sensitive to changes in deep water formation rates and/or
processes. For example, previous studies from the Norwegian Sea found a close correlation between the
composition of benthic foraminiferal assemblages and climate for the past the past 150 kyr – and
identified three main benthic foraminiferal assemblages as “interglacial”, “glacial stadial” and “glacial
interstadial” fauna^{44,19}. The Interglacial assemblage consists of *Cassidulina neoteretis*, *Cassidulina*
*reniforme*, *Melonis barleanus*, and *Islandiella norcrossi*. However, these studies did not examine in
details potential changes in the benthic fauna within the interglacial periods. Our results show that the
Holocene benthic fauna is dominated by *C. neoteretis* and *C. reniforme* constituting together ~60–80 %
with up to ~20% *M. barleanus* (Figure 3c). This faunal composition is similar to the late LIG, but the
relative abundances of these species for the early LIG are distinctly different. During the early LIG, both
*C. neoteretis* and *C. reniforme* constituting <10% of the benthic foraminiferal assemblages with up to
~50% *M. barleanus* (Figure 3c). Although, these differences in the foraminiferal fauna are not well
understood, they point to different bottom water conditions (likely linked to deep water formation
processes and/or rates) compared to the Holocene and late LIG. **Changes in productivity and food**
**availability are unlikely explanations for these changes as infaunal benthic $\delta^{13}\text{C}$ values are the same for**
**both the late and early LIG (see Supplemental Figure 2). The early LIG benthic foraminiferal fauna**
**resembles these of the Younger Dryas stadial (12.8–11.5 ka; ref. 45), when deep water formation in the**
**Nordic Seas is thought to be suppressed⁴⁶.**

Our findings suggest that persistent melting of Arctic sea ice and southward flux of buoyancy
during the Early LIG (~128–124 ka) have altered deep water formation rates in the Nordic Seas and thus
suppressed northward ocean heat transport (i.e., reduced Nordic Seas heat pump). This highlights the
sensitivity of Nordic Seas open ocean convection and regional climate to buoyancy anomalies under
interglacial conditions. Satellite observations indicate that sea-ice cover in the Arctic Ocean has declined
substantially during the past four decades and it is predicted that consistently ice-free summer conditions
will occur by mid-century ~2050 (e.g., refs. 47, 48) with serious implications on climate and ecosystems,
though in unpredictable ways (e.g., refs. 3–8). Our study showcases the complex feedback interactions
between a warming climate and Arctic sea ice and identify the early LIG as a key time interval for data-
model intercomparison efforts to better understand the impacts of a changing cryosphere on regional
and global climate.

Figures and figure captions

Figure 1. Nordic Seas circulation, study area and proxy records for the peak Glacial Maximum of Marine Isotope Stage (MIS) 6, Termination II (deglaciation), MIS 5e comprising the last interglacial LIG, and glacial inception MIS 5d. (a) Map showing major surface (red and white arrows) and bottom (black arrows) currents in the northern North Atlantic and Nordic Seas⁴⁹. Red and white arrows indicate the northward Atlantic surface water inflow and southward polar water outflow, respectively. EGC and EIC refer to the East Greenland and East Icelandic Current, respectively. Circles highlight the location of sediment cores JM11-FI-19PC (black, 1179 m water depth), MD95-2009 (green, 1217 m water depth), LINK16 (blue, 773 m water depth) and ENAM33 (purple circle, 1217 m water depth). The map is modified after ref. 50. **(b)** Benthic foraminiferal $\delta^{18}\text{O}$ from cores JM11-FI-19PC (black, ref. 20), LINK16 (blue, this study), MD95-2009 (green, ref. 23). **(c)** Planktic foraminiferal $\delta^{18}\text{O}$ from cores JM11-FI-19PC (black, ref. 20), LINK16 (blue, this study), MD95-2009 (green, ref. 19). **(d)** Percentage of *Neogloboquadrina pachyderma* from core LINK16 (blue, ref. 25) and core MD95-2009 (green, ref. 19). **(e)** Percentage of planktic foraminiferal species *N. pachyderma* from core ENAM33 (purple, ref. 19). **(f)** Summer solar insolation at 60° N (solid line) and 90° N (dashed line) (ref. 37). The vertical black line indicates the tephra layer 5e-Low/BasIV that is present in the 3 sediment cores.

**Figure 2. Development of sea surface temperature and sea ice in the Norwegian Sea during the**
**penultimate deglaciation and Last Interglacial (LIG).** Sea ice proxies: (a) Relative abundance of
sea-ice indicating diatom species²⁴; (b) Sea ice index $P_{BIP_{25}}$ from two sediment cores (this study); and
(c) Concentration of IP_{25} (a C_{25} Isoprenoid Lipid) from two sediment cores (this study). Temperature
proxies: (d) Relative abundance of warm water-indicating diatom species²⁴; (e) Relative abundance of
cold water-indicating diatom species²⁴; and (f) Relative abundance of the polar planktic foraminiferal
species *Neogloboquadrina pachyderma*^{19,25}. Black, blue and green colors refer to data from sediment
core JM11-FI-19PC, LINK16 and MD95-2009, respectively (see Figure 1a for the core locations).
MIS refers to Marine Isotope Stage.

**Figure 3. Surface and deep ocean conditions during the last two interglacials, the Holocene (10–0**
**ka; left) and the Last Interglacial (128–117 ka; right).** (a) Relative abundance of warm-water
indicating diatom species²⁴ (Hoff et al., 2019). (b) Relative abundance of the polar water planktic
foraminiferal species *Neogloboquadrina pachyderma*^{19, 25}. (c) Relative abundance of the benthic
foraminiferal species *Cassidulina neoteretis* + *Cassidulina reniforme* (solid line and filled circles) and
*Melonis barleeanus* (dashed line and open circles). Black, blue and green colors refer to data from
sediment cores JM11-FI-19PC, LINK16 and MD95-2009, respectively (see Figure 1a for core
locations). Horizontal lines refer to the average Holocene values. Blue arrows refer to an interval in
core LINK16 that is likely impacted by bioturbation (see supplemental Figure 1). MIS stands for
Marine Isotope Stage.

**Figure 4. Freshwater anomalies and their potential sources during the Last Interglacial and**
**penultimate deglaciation.** Salinity proxies **(a)** Relative abundance of the C₃₇ tetra-unsaturated methyl
alkenone (%C_{37:4}); and **(b)** Na/Ca measured in the shells of planktic foraminiferal species
*Neogloboquadrina pachyderma*. Glacier and ice-berg meltwater proxies: **(c)** Ba/Ca measured in *N.*
*pachyderma*; and **(d)** content of Ice Rafted Debris (IRD) counted in >150 µm size fraction¹⁹. **(e)**
Relative abundance of warm-water indicating diatom species²⁴. **(f)** δ¹⁸O measured on *N. pachyderma*.
Black, blue and green colors refer to data from sediment cores JM11-FI-19PC, LINK16 and MD95-
2009, respectively (see Figure 1a for core locations). MIS stands for Marine Isotope Stage.

**References**

- 1. Screen, J. A., & Simmonds, I. (2010). The central role of diminishing sea ice in recent
Arctic temperature amplification. *Nature*, 464(7293), 1334-1337.
- 2. Overland, J., et al. (2019). The urgency of Arctic change. *Polar Science*, 21, 6-13.
- 3. Sévellec, F., Fedorov, A. V., & Liu, W. (2017). Arctic sea-ice decline weakens the Atlantic
meridional overturning circulation. *Nature Climate Change*, 7(8), 604-610.
- 4. Rahmstorf, S., Box, J. E., Feulner, G., Mann, M. E., Robinson, A., Rutherford, S., &
Schaffernicht, E. J. (2015). Exceptional twentieth-century slowdown in Atlantic
Ocean overturning circulation. *Nature climate change*, 5(5), 475-480.
- 5. Dukhovskoy, D. S., et al. (2019). Role of Greenland freshwater anomaly in the recent
freshening of the subpolar North Atlantic. *Journal of Geophysical Research:*
*Oceans*, 124(5), 3333-3360.
- 6. Yashayaev, I., & Loder, J. W. (2017). Further intensification of deep convection in the
Labrador Sea in 2016. *Geophysical Research Letters*, 44(3), 1429-1438.
- 7. Jochumsen, K., et al. (2017). Revised transport estimates of the Denmark S trait
overflow. *Journal of Geophysical Research: Oceans*, 122(4), 3434-3450.
- 8. Lique, C., & Thomas, M. D. (2018). Latitudinal shift of the Atlantic Meridional
Overturning Circulation source regions under a warming climate. *Nature Climate*
*Change*, 8(11), 1013-1020.
- 9. Otto-Bliesner, B. L., Marshall, S. J., Overpeck, J. T., Miller, G. H., Hu, A., & CAPE Last
Interglacial Project members. (2006). Simulating Arctic climate warmth and icefield
retreat in the last interglaciation. *science*, 311(5768), 1751-1753.
- 10. North Greenland Ice Core Project Members (NGRIP Members) (2004), High-resolution
record of Northern Hemisphere climate extending into the last interglacial
period, *Nature*, **431**, 147–151, doi:10.1038/nature02805.
- 11. EPICA Community members (2004). Eight glacial cycles from an Antarctic ice
core. *Nature*, 2004, 429.6992: 623-628.
- 12. Stein, R., Fahl, K., Gierz, P., Niessen, F., & Lohmann, G. (2017). Arctic Ocean sea ice
cover during the penultimate glacial and the last interglacial. *Nature*
*communications*, 8(1), 373.
- 13. Vermassen, F., et al. (2023). A seasonally ice-free Arctic Ocean during the Last
Interglacial. *Nature Geoscience*, 16(8), 723-729.
- 14. Razmjooei, M. J., et al. (2023). Revision of the Quaternary calcareous nannofossil
biochronology of Arctic Ocean sediments. *Quaternary Science Reviews*, 321,
108382.
- 15. Hillaire-Marcel, C., et al. (2017). A new chronology of late Quaternary sequences from
the central Arctic Ocean based on “extinction ages” of their excesses in 231Pa and
230Th. *Geochemistry, Geophysics, Geosystems*, 18(12), 4573-4585.
- 16. Bauch, H. A., Erlenkeuser, H., Fahl, K., Spielhagen, R. F., Weinelt, M. S., Andrulleit, H.,
& Henrich, R. (1999). Evidence for a steeper Eemian than Holocene sea surface
temperature gradient between Arctic and sub-Arctic regions. *Palaeogeography,*
*Palaeoclimatology, Palaeoecology*, 145(1-3), 95-117.
- 17. Van Nieuwenhove, N., & Bauch, H. A. (2008). Last interglacial (MIS 5e) surface water
conditions at the Vøring Plateau (Norwegian Sea), based on dinoflagellate
cysts. *Polar Research*, 27(2), 175-186.

- 18. Risebrobakken, B., Dokken, T., & Jansen, E. (2005). Extent and variability of the
Meridional Atlantic Circulation in the Eastern Nordic Seas during Marine Isotope
Stage 5 and its influence on the inception of the last Glacial. *Geophysical*
*Monograph Series*, 158, 323-339.
- 19. Rasmussen, T. L., Thomsen, E., Kuijpers, A., & Wastegård, S. (2003). Late warming and
early cooling of the sea surface in the Nordic seas during MIS 5e (Eemian
Interglacial). *Quaternary Science Reviews*, 22(8-9), 809-821.
- 20. Ezat, M. M., Rasmussen, T. L., & Groeneveld, J. (2016). Reconstruction of hydrographic
changes in the southern Norwegian Sea during the past 135 kyr and the impact of
different foraminiferal Mg/Ca cleaning protocols. *Geochemistry, Geophysics,*
*Geosystems*, 17(8), 3420-3436.
- 21. Govin, A., et al. (2012). Persistent influence of ice sheet melting on high northern latitude
climate during the early Last Interglacial. *Climate of the Past*, 8(2), 483-507.
- 22. Capron, E., et al. (2014). Temporal and spatial structure of multi-millennial temperature
changes at high latitudes during the Last Interglacial. *Quaternary Science*
*Reviews*, 103, 116-133.
- 23. Van Nieuwenhove, N., Bauch, H. A., Eynaud, F., Kandiano, E., Cortijo, E., & Turon, J. L.
(2011). Evidence for delayed poleward expansion of North Atlantic surface waters
during the last interglacial (MIS 5e). *Quaternary Science Reviews*, 30(7-8), 934-946.
- 24. Hoff, U., Rasmussen, T. L., Meyer, H., Koç, N., & Hansen, J. (2019). Palaeoceanographic
reconstruction of surface-ocean changes in the southern Norwegian Sea for the last~
130,000 years based on diatoms and with comparison to foraminiferal
records. *Palaeogeography, palaeoclimatology, palaeoecology*, 524, 150-165.
- 25. Abbott, P. M., Austin, W. E., Davies, S. M., Pearce, N. J. G., Rasmussen, T. L.,
Wastegård, S., & Brendryen, J. (2014). Re-evaluation and extension of the Marine
Isotope Stage 5 tephrostratigraphy of the Faroe Islands region: The cryptotephra
record. *Palaeogeography, Palaeoclimatology, Palaeoecology*, 409, 153-168.
- 26. Kucera, M., et al. (2005). Reconstruction of sea-surface temperatures from assemblages of
planktonic foraminifera: multi-technique approach based on geographically
constrained calibration data sets and its application to glacial Atlantic and Pacific
Oceans. *Quaternary Science Reviews*, 24(7-9), 951-998.
- 27. Hoff, U., Rasmussen, T. L., Stein, R., Ezat, M. M., & Fahl, K. (2016). Sea ice and
millennial-scale climate variability in the Nordic seas 90 kyr ago to present. *Nature*
*communications*, 7(1), 12247.
- 28. Andersson, C., Pausata, F. S., Jansen, E., Risebrobakken, B., & Telford, R. J. (2010).
Holocene trends in the foraminifer record from the Norwegian Sea and the North
Atlantic Ocean. *Climate of the Past*, 6(2), 179-193.
- 29. Zamelczyk, K., Rasmussen, T. L., Husum, K., Godtliebsen, F., & Hald, M. (2014).
Surface water conditions and calcium carbonate preservation in the Fram Strait
during marine isotope stage 2, 28.8–15.4 kyr. *Paleoceanography*, 29(1), 1-12.
- 30. Griggs, A.J., Davies, S.M., Abbott, P.M., Rasmussen, T.L., Palmer, A.P., 2014.
Optimising the use of marine tephrochronology in the North Atlantic: a detailed
investigation of the Faroe Marine Ash Zones II, III and IV. *Quaternary Science*
*Reviews* 106, 122-139.
- 31. Steinsland, K., Grant, D. M., Ninnemann, U. S., Fahl, K., Stein, R., & De Schepper, S.
(2023). Sea ice variability in the North Atlantic subpolar gyre throughout the last
interglacial. *Quaternary Science Reviews*, 313, 108198.

- 32. Brown, T. A., Belt, S. T., Tatarek, A., & Mundy, C. J. (2014). Source identification of the
Arctic sea ice proxy IP25. *Nature Communications*, 5(1), 4197.
- 33. Belt, S. T., Massé, G., Rowland, S. J., Poulin, M., Michel, C., & LeBlanc, B. (2007). A
novel chemical fossil of palaeo sea ice: IP25. *Organic Geochemistry*, 38(1), 16-27.
- 34. Müller, J., Wagner, A., Fahl, K., Stein, R., Prange, M., & Lohmann, G. (2011). Towards
quantitative sea ice reconstructions in the northern North Atlantic: A combined
biomarker and numerical modelling approach. *Earth and Planetary Science
Letters*, 306(3-4), 137-148.
- 35. Ruddiman, W. F., McIntyre, A., Niebler-Hunt, V., & Durazzi, J. T. (1980). Oceanic
evidence for the mechanism of rapid northern hemisphere glaciation. *Quaternary
Research*, 13(1), 33-64.
- 36. Risebrobakken, B., Dokken, T., Otterå, O. H., Jansen, E., Gao, Y., & Drange, H. (2007).
Inception of the Northern European ice sheet due to contrasting ocean and insolation
forcing. *Quaternary Research*, 67(1), 128-135.
- 37. Berger, A. (1978). Long-term variations of daily insolation and Quaternary climatic
changes. *Journal of Atmospheric Sciences*, 35(12), 2362-2367.
- 38. Guay, C. K., & Falkner, K. K. (1997). Barium as a tracer of Arctic halocline and river
waters. *Deep Sea Research Part II: Topical Studies in Oceanography*, 44(8), 1543-
1569.
- 39. Hönisch, B., et al. (2011). Planktic foraminifers as recorders of seawater Ba/Ca. *Marine
Micropaleontology*, 79(1-2), 52-57.
- 40. Ekwurzel, B., Schlosser, P., Mortlock, R. A., Fairbanks, R. G., & Swift, J. H. (2001).
River runoff, sea ice meltwater, and Pacific water distribution and mean residence
591 times in the Arctic Ocean. *Journal of Geophysical Research: Oceans*, 106(C5),
9075-9092.
- 41. Bauch, D., Schlosser, P., & Fairbanks, R. G. (1995). Freshwater balance and the sources
of deep and bottom waters in the Arctic Ocean inferred from the distribution of
H₂¹⁸O. *Progress in Oceanography*, 35(1), 53-80.
- 42. Bertlich, J., et al. (2018). Salinity control on Na incorporation into calcite tests of the
planktonic foraminifera *Trilobatus sacculifer*—evidence from culture experiments and
surface sediments. *Biogeosciences*, 15(20), 5991-6018.
- 43. Wang, K. J., et al. (2021). Group 2i Isochrysidales produce characteristic alkenones
reflecting sea ice distribution. *Nature Communications*, 12(1), 15.
- 44. Rasmussen, T. L., Balbon, E., Thomsen, E., Labeyrie, L., & Van Weering, T. C. (1999).
Climate records and changes in deep outflow from the Norwegian Sea~ 150–55
603 ka. *Terra Nova*, 11(2-3), 61-66.
- 45. Rasmussen, T. L., Thomsen, E., Labeyrie, L., & van Weering, T. C. (1996). Circulation
changes in the Faeroe-Shetland Channel correlating with cold events during the last
glacial period (58–10 ka). *Geology*, 24(10), 937-940.
- 46. Muschitiello, F., et al. (2019). Deep-water circulation changes lead North Atlantic climate
during deglaciation. *Nature communications*, 10(1), 1272.
- 47. Cavalieri, D. J., Parkinson, C. L., Gloersen, P., Comiso, J. C., & Zwally, H. J. (1999).
Deriving long-term time series of sea ice cover from satellite passive-microwave
multisensor data sets. *Journal of Geophysical Research: Oceans*, 104(C7), 15803-
15814.
- 48. Jahn, A., Holland, M. M., & Kay, J. E. (2024). Projections of an ice-free Arctic
Ocean. *Nature Reviews Earth & Environment*, 1-13.

- 49. Hansen, B., & Østerhus, S. (2000). North atlantic–nordic seas exchanges. *Progress in*
*oceanography*, 45(2), 109-208.
- 50. Ezat, M. M., Rasmussen, T. L., & Groeneveld, J. (2014). Persistent intermediate water
warming during cold stadials in the southeastern Nordic seas during the past 65
ky. *Geology*, 42(8), 663-666.
- 51. Veres, D., Bazin, L., Landais, A., Toyé Mahamadou Kele, H., Lemieux-Dudon, B.,
Parrenin, F., Martinerie, P., Blayo, E., Blunier, T., Capron, E., Chappellaz, J.,
Rasmussen, S. O., Severi, M., Svensson, A., Vinther, B., and Wolff, E. W.: The
Antarctic ice core chronology (AICC2012): an optimized multi-parameter and multi-
site dating approach for the last 120 thousand years, *Clim. Past*, 9, 1733–1748,
<https://doi.org/10.5194/cp-9-1733-2013>, 2013.
- 52. Bond, G., Broecker, W., Johnsen, S., McManus, J., Labeyrie, L., Jouzel, J., & Bonani, G.
(1993). Correlations between climate records from North Atlantic sediments and
Greenland ice. *Nature*, 365(6442), 143-147.
- 53. Chappellaz, J., Blunier, T., Raynaud, D., Barnola, J. M., Schwander, J., & Stauffert, B.
(1993). Synchronous changes in atmospheric CH₄ and Greenland climate between
40 and 8 kyr BP. *Nature*, 366(6454), 443-445.
- 54. Boon, J. J., W. I. C. Rijpstra, F. de Lange, J. De Leeuw, M. YOSHIOKA, and Y.
SHIMIZU. 1979. Black Sea sterol—a molecular fossil for dinoflagellate blooms.
*Nature* 277:125-127.
- 55. Volkman, J. K. 1986. A review of sterol markers for marine and terrigenous organic
matter. *Organic Geochemistry* 9:83-99.
- 56. Fahl, K., and R. Stein. 2012. Modern seasonal variability and deglacial/Holocene change
of central Arctic Ocean sea-ice cover: new insights from biomarker proxy records.
*Earth and Planetary Science Letters* 351:123-133.
- 57. Bard, E., Rostek, F., Turon, J.L., Gendreau, S., 2000. Hydrological impact of Heinrich
events in the subtropical Northeast Atlantic. *Science* 289, 1321–1324
- 58. Ezat, M. M., Rasmussen, T. L., Thornalley, D. J., Olsen, J., Skinner, L. C., Hönisch, B., &
Groeneveld, J. (2017). Ventilation history of Nordic Seas overflows during the last
(de) glacial period revealed by species-specific benthic foraminiferal ¹⁴C
dates. *Paleoceanography*, 32(2), 172-181.
- 59. Martin, P. A., & Lea, D. W. (2002). A simple evaluation of cleaning procedures on fossil
benthic foraminiferal Mg/Ca. *Geochemistry, Geophysics, Geosystems*, 3(10), 1-8.
- 60. Mackensen, A., & Schmiedl, G. (2019). Stable carbon isotopes in paleoceanography:
atmosphere, oceans, and sediments. *Earth-Science Reviews*, 197, 102893.
- 61. Duplessy, J. C., Shackleton, N. J., Fairbanks, R. G., Labeyrie, L., Oppo, D., & Kallel, N.
(1988). Deepwater source variations during the last climatic cycle and their impact
on the global deepwater circulation. *Paleoceanography*, 3(3), 343-360.

**Acknowledgement.** We thank N. El bani Altuna, T. Dahl, I. Hald, K. Monsen, M. Lindgren, B.
Honish, J. Ruprecht, L. Pena, K. Esswein, and W. Luttmer for laboratory support. We also thank E.
Capron for sharing data from sediment core MD95-2009. This study is financed by a starting grant
from the Tromsø Forskningsstiftelse to to M.M.E, project number A31720. The research also received
support from the Research Council of Norway and the Co-funding of Regional, National, and
International Programmes (COFUND)—Marie Skłodowska-Curie Actions under the EU Seventh
Framework Programme (FP7), project number 274429.

**Author Contributions.** M.M.E conceived and designed the study, performed most of the
foraminiferal geochemical analyses and wrote the manuscript. T.L.R provided the isotope data and
benthic foraminiferal assemblages from core LINK16. K.F conducted the biomarker analyses,
evaluation, and quality control. All authors contributed to the discussions and the final draft of the
manuscript.

**Data Availability.** Upon the acceptance for publication, all new data presented in this article will be
made available at UiT Open Research Data Repository.

**Competing interests.** The authors declare no competing interests.

**Methods**

**1. Chronology**

[revised manuscript text omitted]

Supplemental Materials

Supplemental Figure 1. Bioturbation event in Core LINK16 (blue-highlighted). (a) Number of basaltic shards per gram counted in the >100 μm in core LINK16. (b) Relative abundance of the benthic foraminiferal species *Cassidulina reniforme*. (c) Relative abundance of the polar planktic foraminiferal species *Neogloboquadrina pachyderma* in cores LINK16 (ref. 25; blue) and MD95-2009 (ref. 19; green). The blue-highlighted area marks an interval in core LINK16 where is a large amplitude, short-term change in planktic and benthic foraminiferal assemblages at 124.5 ka – in particular an increase in the subpolar planktic foraminiferal species. These changes are not recorded in the other nearby Norwegian Sea cores JM11-FI-19PC and MD95-2009. At the same interval in LINK16, the concentration of basaltic shards decreased suggesting that this interval could have been affected by bioturbation from the red-highlighted interval. It is possible that a burrow has brought sediments that are rich in the subpolar planktic foraminifera and poor in basaltic grains from late LIG sediments (the red-highlighted interval). The bioturbation effects are clear in the basaltic shard counts, benthic and planktic fauna because of the high contrast in these parameters between the early and late LIG parts. The bioturbation effects are not clear in other proxies such as foraminiferal $\delta^{18}\text{O}$ and IP_{25} because these are not significantly different between the two intervals (Figure 1b, c; Figure 2c). This highlights the importance of the multi-core investigation and needed caution with single core-based studies.

Supplemental Figure 2. $\delta^{13}\text{C}$ measured in the infaunal benthic foraminiferal species *M. barleanus* in sediment cores LINK16 (blue) and JM11-FI-19PC (black). The grey box highlights the whole LIG interval. Infaunal benthic $\delta^{13}\text{C}$ responds to changes in the flux of organic matter and/or oxygen availability within their environment (the upper few centimetres in the sediments; e.g., ref. 60). Stable infaunal benthic $\delta^{13}\text{C}$ can be indicative on stable food supply to the seabed. It is noteworthy that epifaunal benthic $\delta^{13}\text{C}$ is often used as an indicator to changes in deep water mass sourcing and ventilation (e.g., ref. 61), but we could not find epifaunal benthic foraminifera in the early LIG interval.

**Supplemental Figure 3. Age models for sediment cores LINK16 and JM11-FI-19PC. (a)**
 Percentage of *N. pachyderma* from core LINK16 (blue, ref. 25) and core MD95-2009 (green, ref. 19).
 **(b)** Benthic foraminiferal $\delta^{18}\text{O}$ from cores JM11-FI-19PC (black, ref. 20), LINK16 (blue, this study),
 MD95-2009 (green, ref. 23). **(c)** Planktic foraminiferal $\delta^{18}\text{O}$ from cores JM-FI-19PC (black, ref. 20),
 LINK16 (blue, this study), MD95-2009 (green, ref. 19). The solid vertical lines refer to the main four
 tie points used in transferring the age model of MD95-2009 (ref. 22) to sediment cores LINK 16 and
 JM-FI-19PC. The dashed line refers to an additional age marker for the bottom of core JM-FI-19PC.
